# ASYNCHRONOUS MESSAGE PASSING: A NEW FRAMEWORK FOR LEARNING IN GRAPHS

## ABSTRACT

This paper studies asynchronous message passing (AMP), a new framework for applying neural networks to graphs. Existing graph neural networks (GNNs) use the message passing framework which is based on the synchronous distributed computing model. In traditional GNNs, nodes aggregate their neighbors in each round, which causes problems such as oversmoothing and expressiveness limitations. On the other hand, our AMP framework is based on the *asynchronous* model, where nodes react to messages of their neighbors individually. We prove: (i) AMP is at least as powerful as the message passing framework, (ii) AMP is more powerful than the $1-$WL test for graph isomorphism, an important benchmark for message passing GNNs, and (iii) in theory AMP can even separate any pair of graphs and compute graph isomorphism. We experimentally validate the findings on AMP's expressiveness, and show that AMP might be better suited to propagate messages over large distances in graphs. We also demonstrate that AMP performs well on several graph classification benchmarks.

## 1 INTRODUCTION

Graph Neural Networks (GNNs) have become the de-facto standard model for applying neural networks to graphs in many domains (Bian et al., 2020; Gilmer et al., 2017; Hamilton et al., 2017; Jumper et al., 2021; Kipf & Welling, 2017; Veličković et al., 2018; Wu et al., 2020). Internally, nodes in GNNs use the message passing framework, i.e., nodes communicate with their neighboring nodes for multiple synchronous rounds. We believe that this style of communication is not always ideal. In GNNs, all nodes speak concurrently, and a node does not listen to individual neighbors but only to an aggregated message of all neighbors.

In contrast, humans politely listen when a neighbor speaks, then decide whether the information was relevant, and what information to pass on. The way humans communicate is in line with the asynchronous communication model (Peleg, 2000). In the asynchronous model, nodes do not communicate concurrently. In fact, a node only acts when it receives a message (or when initialized). If a node receives a new message from one of its neighbors, it updates its state, and then potentially sends a message on its own. This allows nodes to listen to individual neighbors and not only to aggregations. Figure 1 illustrates how this interaction can play out.

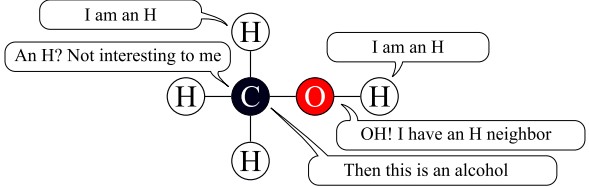

Figure 1: Detection of an alcohol (a C atom with an OH group) with AMP. H atoms learned to initially send a message to their neighbors. Every node can choose to ignore the message or react to it. The C atom is not interested in H neighbors and discards the message. On the other hand, the O atom reacts and sends a message on its own. This message is now relevant to the C atom.

We make the following contributions.

- We introduce AMP, a new framework for learning neural architectures in graphs. Instead of nodes acting synchronously in rounds, nodes in AMP interact asynchronously by exchanging and reacting to individual messages.
- We theoretically examine AMP and prove that the AMP framework is at least as powerful as the synchronous message passing network. Furthermore, we show that AMP can separate graphs beyond the $1-$WL test; conceptually AMP can solve graph isomorphism.
- We examine how AMP can transmit information to far-away nodes. Since AMP handles messages individually and is not limited by the number of communication rounds, AMP can combat the underreaching, oversmoothing, and oversquashing problems that traditional GNNs encounter when propagating information over long distances (many layers).
- We run experiments on (i) established GNN expressiveness benchmarks to demonstrate that AMP outperforms all existing methods in distinguishing graphs beyond the $1-$WL algorithm. We introduce (ii) synthetic datasets to show that AMP is well suited to propagate information over large distances. Finally, we study (iii) established graph classification benchmarks to show that AMP performs comparably to existing GNNs.

## 2 RELATED WORK

Apart from some domains with special graph families (for example processing directed acyclic graphs sequentially (Amizadeh et al., 2018)), virtually all GNNs follow the synchronous message passing framework of distributed computing, first suggested by Gilmer et al. (2017) and Battaglia et al. (2018). The underlying idea is that nodes have an embedding and operate in rounds. In each round, every node computes a message and passes the message to every adjacent node. Then, every node aggregates the messages it receives and uses the aggregation to update its embedding. There exist variations of this framework. For example, edges can also have embeddings, or one can add a global sharing node to allow far away nodes to directly share information (Battaglia et al., 2018). Following the initial work of Scarselli et al. (2008), different implementations for the individual steps in the message passing framework exist, e.g., (Brody et al., 2022; Hamilton et al., 2017; Kipf & Welling, 2017; Niepert et al., 2016; Veličković et al., 2018; Xu et al., 2018; 2019). However, these GNN architectures all experience common problems:

**Oversmoothing.** A problem that quickly emerged with GNNs is that we cannot have many GNN layers (Li et al., 2019; 2018). Each layer averages and hence smooths the neighborhood information and the node's features. This effect leads to features converging after some layers (Oono & Suzuki, 2020), which is known as the oversmoothing problem. Several works address the oversmoothing problem, for example by sampling nodes and edges to use in message passing (Feng et al., 2020; Hasanzadeh et al., 2020; Rong et al., 2020), leveraging skip connections (Chen et al., 2020b; Xu et al., 2018), or additional regularization terms (Chen et al., 2020a; Zhao & Akoglu, 2020; Zhou et al., 2020). Thanks to its asynchrony, AMP does not average over neighborhood messages. This helps preserving the identity of individual messages and makes AMP more resilient against the oversmoothing problem.

**Underreaching.** Using normal GNN layers, a GNN with $k$ layers only learns about nodes at most $k$ hops away. A node cannot act correctly if it would need information that is $k+1$ hops away. This problem is called underreaching (Barceló et al., 2020). There exist countermeasures, for example, having a global exchange of features (Gilmer et al., 2017; Wu et al., 2021) or spreading information using diffusion processes (Klicpera et al., 2019; Scarselli et al., 2008). Methods that help against oversmoothing are usually also applied against underreaching since we can use more layers and increase the neighborhood size. In AMP, because of asynchrony, some nodes can be involved in the computation/communication much more often than others; this helps AMP to gather information from further away, which is a countermeasure against underreaching.

**Oversquashing.** In many graphs, the size of $k-$hop neighborhoods grows substantially with $k$. This requires squashing more and more information into a node embedding of static size. Eventually, this leads to the congestion problem (too much information having to pass through a bottleneck) that is well known in distributed computing (e.g. (Sarma et al., 2012)) and goes by the name of oversquashing for GNNs (Alon & Yahav, 2021; Topping et al., 2022). One approach to solve oversquashing is

introducing additional edges that function as shortcuts to non-direct neighbors (Brüel-Gabrielsson et al., 2022). Dropping-based methods (Feng et al., 2020; Hasanzadeh et al., 2020; Rong et al., 2020) that help against oversmoothing can also reduce oversquashing by reducing the size of the neighborhoods. AMP is more resilient than synchronous GNNs. The GNN aggregation step is prone to bury a relevant message in background noise, whereas AMP will see this message in isolation.

$1-$**WL Limit.** Xu et al. (2019) and Morris et al. (2019) show that GNNs are limited in their expressiveness by the $1-$Weisfeiler-Lehman test ($1-$WL), a heuristic algorithm to evaluate graph isomorphism (Shervashidze et al., 2011). However, there exist simple structures that the $1-$WL test cannot distinguish that we want to detect with GNNs (Garg et al., 2020). Therefore, several augmentations to GNNs exist that include additional features, such as ports, IDs, or angles between edges for chemistry datasets (Gasteiger et al., 2020; Loukas, 2020; Sato et al., 2019; 2021), run multiple rounds over slight perturbations of the same graph (Bevilacqua et al., 2022; Papp et al., 2021; Vignac et al., 2020), or use higher-order information (Chen et al., 2019; Maron et al., 2019; Morris et al., 2019). Since in AMP, nodes do not act at the same time, they can be distinguished more easily. We show that AMP can handle $1-$WL and higher order WL, in principle even graph isomorphism.

We believe these four classic GNN problems have their root cause in the *synchronous aggregation* of (almost) all neighbors. The aggregation smooths the neighborhood features, and nodes are exposed to all the information in their increasing $k - hop$ neighborhood. This limits the number of rounds, causing underreaching. If nodes with identical neighbors act synchronously, they stay identical and within $1-$WL expressiveness. A recent work by Schaefer et al. (2022) employs some asynchrony to decide which nodes in the graph require processing but node updates are still done in a synchronous fashion. We believe that only full asynchrony with individual processing of messages can yield all the advantages.

Interestingly, a similar observation was made in the field of probabilistic graphical models. Elidan et al. (2012) discuss that one method, called belief propagation, often does not converge or converges slowly when every node in the graphical model communicates synchronously. The authors then show that asynchronous computation, processing one node at a time can lead to better results. Knoll et al. (2015) and Aksenov et al. (2020) build on these results and improve the asynchronous model with improved message scheduling.

There are two main approaches for distributed computing (Peleg, 2000; Wattenhofer, 2020). In the synchronous approach, all nodes operate in synchronous rounds, and in each round every node sends a message to every neighbor. Sato et al. (2019) and Loukas (2020) show that current graph neural networks using message passing as in Gilmer et al. (2017) follow this approach. The antipodal paradigm in distributed computing is the asynchronous model Peleg (2000). In the asynchronous model, nodes do not act at the same time. Instead, nodes act on receiving a single message. If receiving a message, a node may change its internal state and potentially create new messages itself. Currently, all GNN approaches follow the synchronous message passing model. In this paper, we want to explore the efficacy of asynchronous message passing. So we ask a natural question: Is it possible, and is it worthwhile for GNNs to handle messages asynchronously and individually?

## 3   THE ASYNCHRONOUS FRAMEWORK

We outline the asynchronous framework through the pseudo algorithm in Algorithm 1. The algorithm describes how AMP computes representations for each node in a given graph $G$ and node features $\boldsymbol{X}$. AMP treats every message individually, and as such (in principle) no information is lost in aggregation. However, asynchronous messages can be scheduled in different ways, which is a challenge for the learning process. Generally messages incur a delay before they arrive. We track all messages that are sent but not yet received with their delay in a priority heap. From here, messages are processed on at a time. AMP delivers the message to the receiving node which executes three steps: (i) The receiving node can decide that the message is not interesting and discard it; (ii) The receiving node updates its state and computes a new message; (iii) The message is sent to each neighbor, by sending the messages with a delay sampled from a distribution $\mathcal{D}$.

There are three major turning knobs in the AMP framework. First, we can vary heap initialization. For example, we can send an initial message to just one node (e.g., when finding shortest paths from this node), send an initial message to all nodes, or execute one run for each node, where this node is

the sole starting node. Second, we can choose from different delay distributions $\mathcal{D}$: The two main cases are constant versus random delays. Third, we can choose how to implement the three neural blocks: (i) the reaction block $\rho$ whether to discard a message; (ii) the state transition function $\delta$; and (iii) the message function $\mu$.

---

**Algorithm 1:** AMP representation learning with a single run and a single starting node.
**Input**: Graph $G$, node features matrix $mX$, bool poolNodes
**Output**: Graph representation if poolNodes, otherwise node representations.

```
1  heap = MinHeap()      # Triples (delay, message, node), sorted by delay
2  receivedCount = 0                        # Safeguard against infinite loop
3  initializeHeap()          # e.g., a message to all node, one single node
4  while not heap.empty() and receivedCount < k do              # for constant k
5      delay, message, receiver = heap.extractMin()
6      if ρ(state[receiver], message) then              # ρ does not discard message
7          receivedCount++
8          state[receiver] = δ(state[receiver], message)           # update node state
9          newMessage = μ(state[receiver], message)
10         for v ∈ G.neighbors(receiver) do
11             heap.insert(delay + d∼ 𝒟, newMessage, v)

    # At this point, states contains a representation per node
12 H = StackToMatrix(states)              # Combine node states to matrix
13 if poolNodes then
14     H = pool(H)                              # Pool for graph classification.
    # Otherwise use H as is for node classification.
15 return H
```

---

## 3.1 AN AMP EXAMPLE RUN

We want to illustrate the pseudocode of Algorithm 1 at an example. We are picking up the example graph from Figure 1 in the introduction. This time we make communication follow the AMP framework closely, each frame depicts one iteration of the while loop. The left side of each frame shows the current status of the heap of unreceived messages. The top message in the heap is processed. In our example, node $C$ receives the initial mesage. In the top left picture, $C$ reacts to the starting message and sends one message to each of its neighbors. The first one to receive this message is the $H$ on the top in the second frame and replies. Next, the $O$ atom receives the message from $C$ in the top right frame. As part of the state update this $O$ can remember it received a message from a $C$. Since COH groups form alcohols, $O$ now needs to find out if its other neighbor is an $H$. However, the next received message in the bottom left frame is $C$ receiving the reply from $H$. This does not give any important information to $C$ and it discards the message. In the next frame, the rightmost $H$ receives the message from $O$. This $H$ already knows that there is a COH group. In the next frame the $O$ also learns. At this point the while loop keeps iterating until some point. This example is a graph classification problem, so the states of $O$ and the right $H$ can contribute to classifying the graph as alcohol, while other nodes might not know. In Appendix A, we give some more intuition about AMP by comparing it to sequence-to-sequence models from NLP and show some concrete example architectures, i.e., how one could realize $\rho$, $\mu$, and $\delta$.

## 3.2 COMPARING AMP TO GNNS

**Complexity.** Let analyze complexity on the directed graph $G$ with $n$ nodes. We compare the complexity of a GNN with $l$ layers and AMP with a limit of $k$ messages. These are usually constant size so we will skip them in our analysis. Computing messages to send and node updates is usually done through constant-sized neural functions. Therefore, we measure complexity by the total number of messages and node updates. For this analysis we investigate GNNs with a fixed number of layers (unlike IterGNN(Tang et al., 2020)) and AMP variants without message discarding (unlike AMP-iter)

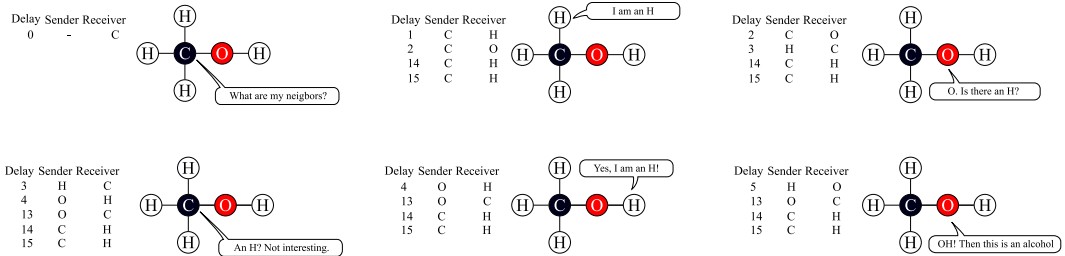

Figure 2: Step-by-step example of AMP on the detection of a COH group. The Figure shows six frames, each showing the processing of a different message. The left hand side of each frame shows the current message queue, while the right hand side show the recipient reacts to the current message. The current message is the top message on the left.

To compute a forward pass in the GNN every layer computes one message per edge and one node update per node. If $\Delta_{max}$ is the maximum degree of a node, we compute $O(n\Delta_{max})$ many messages and $O(n)$ node updates. Some GNNs, such as ESAN (Bevilacqua et al., 2022), computes $|S|$ many runs, where $|S|$ is possibly $O(n)$. The total complexity of ESAN is $O(|S|n\Delta_{max})$. On the other hand DropGNN (Papp et al., 2021), performs $r$ many runs with a complexity of $O(rn\Delta_{max})$.

In a single AMP run, we compute $k$ messages and each message leads to one node update. As part of the node update a node sends up to $\Delta_{max}$ many messages to its neighbors. However, to produce expressive representations, we usually do $O(n)$ (comparable to SMP(Vignac et al., 2020) or (Bevilacqua et al., 2022)) such runs which originate form different nodes. Overall, AMP sends $O(n\Delta_{max})$ messages and does $O(n)$ node updates.

**Expressiveness.** AMP and GNNs both compute latent representations for nodes that we can use for learning a downstream task through a readout function. Instead of running AMP as in Algorithm 1, we also could have used a message passing graph neural network to learn a representation of the nodes or graph. The question lends itself when we should use AMP and when a GNN. For the GNN, we assume a message passing neural network following the framework of Gilmer et al. (2017) that operates in rounds and computes the following in each round: Nodes compute a message based on their current representation and send these to all their neighbors. Then, nodes update their state based on the aggregation of the received messages and their previous state.

$$m_u = \texttt{MESSAGE}(h_u)$$
$$h_{u+1} = \texttt{UPDATE}(h_u, \sum_{v \in N(u)} m_v)$$

Our first main result is that AMP can compute every node representation that the message passing framework can compute.

**Theorem 3.1.** *Let $G$ be a graph with node features $X$. Let $\mathcal{G}$ be a message passing neural network that produces node representations $z_v^l$ for every node $v$ at every layer $l$. We can create an AMP network $\mathcal{A}_\mathcal{G}$ that, for every $G, \boldsymbol{X}, l, v$ can compute the exact same representations that $\mathcal{G}$ does. AMP also consciously reaches these representations and use them. We say that $\mathcal{A}_\mathcal{G}$ simulates $\mathcal{G}$ since we could plug AMP into any task requiring the message passing GNN and use AMP in its stead.*

We attach the proof in Appendix B.1. We start from the $\alpha$ synchronizer from Awerbuch (1985) and show that we can create a similar synchronizer in the framework of Algorithm 1. While we did not consider edge features and or embeddings here, the proof can easily be extended to allow these as well. Similarly, we can include a global node. Therefore AMP can also simulate GNNs captured by the framework of Battaglia et al. (2018). Similarly AMP can also simulate GNNs that use other permutation invariant aggregation functions.

## 4 EXPRESSIVENESS OF AMP

### 4.1 GENERAL ANALYSIS OF AMP'S EXPRESSIVENESS

**Lemma 4.1.** *AMP is at least as powerful as the $1-WL$ test.*

*Proof.* This observation follows from Theorem 3.1. GIN (Xu et al., 2019) is a message passing neural network that is exactly as powerful as the $1-WL$ test. This means that GIN can compute representations to separate $1-WL$ separable graphs. We now know that AMP can simulate GIN and compute the same representations and use them to separate the same graphs. □

**Lemma 4.2.** *Let $v$ be a node in a cycle of length $k$ starting an AMP execution. All nodes in the cycle can learn that they are in a cycle and its length $k$.*

We supply the proof in Section B.2. Basically, AMP searches from both ends of the cycle and counts until the searches meet. The cycle length is the sum of the two counts. As a consequence, AMP can separate the two constructions from Garg et al. (2020) that we depict in Figure 3b by deciding whether or not the graph has a cycle of length 8. By learning if the graph has a $3-$ cycle using only one blue node, AMP can also separate the graph in Figure 3c.

**Lemma 4.3.** *Let $v$ be a node in $k + 1$ clique with nodes $w_1, w_2, \ldots w_k$ starting an AMP execution. All nodes can learn that they are in a $k + 1$ clique.*

We supply the proof in Section B.3. In the proof, the nodes iteratively find out that they are in $1, 2, \ldots k + 1$ cliques by waiting for $k$ confirmations at each step. As a consequence, AMP can separate the rook and shrikande graphs (rook graphs have a 4-clique while shrikande graphs do not), which are not even separable by $3-WL$ (Chen et al., 2019).

**Corollary 4.4.** *AMP can separate graphs that are harder than the $1WL-$ test.*

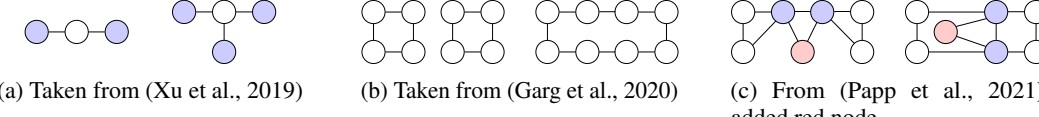

(a) Taken from (Xu et al., 2019)    (b) Taken from (Garg et al., 2020)    (c) From (Papp et al., 2021), added red node.

Figure 3: a) Graphs that cannot be distinguished by $1-WL$ GNNs using `max` aggregation. However, the both graphs have a different number of blue nodes responding when the central white node sends a message. AMP handles messages separately and can separate the graphs. Graphs in b) and c) cannot be separated by any $1WL-$ GNN but by AMP which can count cycle lengths (Lemma 4.2).

### 4.2 AMP WITH RANDOM MESSAGE DELAYS

Let us revisit the example in Figure 3c and suppose we start on the red node. Therefore, we cannot use Lemma 4.2 since the node is not part of the cycle. In case of constant message delays, both blue neighbors receive messages at the same time. Then all white nodes receive messages at the same time and ultimatively we cannot separate the graphs. On the other hand, if delays are random, one blue node can receive and send a message to the other blue node while that node has not yet received a message from the red node. This "slower" blue node will not act, and the "faster" blue node checks for a $3-$ cycle. This example illustrates that delays can act as a symmetry breaker and benefit expressiveness. Let us understand the benefits in a star graph:

**Lemma 4.5.** *AMP with random message delays can create unique identifiers (IDs) for every node in a star graph.*

We provide a proof Lemma 4.5 in Appendix B.4. In principle, the central node auctions IDs to the star nodes and restarts offering an ID when two or more nodes claim the ID at the same time. We can generalize this proof from star graphs to arbitrary connected graphs. We start with an arbitrary node that assigns IDs in its $1-$hop neighborhood (a star graph). Next, the node with the next-highest ID repeats the process — skipping nodes that already have an ID. Eventually every node is adjacent to a center node and received an ID. The algorithm stops when every node tried assigning IDs and finding

a successor is not possible. We can further extend this scheme to unconnected graphs by adding a master node that is connected to every node. We can conclude:

**Corollary 4.6.** *AMP with random message delays can create IDs in arbitrary graphs.*

We can now define the expressiveness of AMP with random message delays.

**Theorem 4.7.** *Given two connected graphs $G_1$, $G_2$ that we connect via a node $u$ to a graph $G$ and diameter $\delta_G$, AMP with random message delays that has a width in $O(\frac{n}{2}!)$ can solve graph isomorphism for any graphs $G_1$, $G_2$ simulating an appropriate GNN for $\delta_G$ many rounds.*

We proof this theorem in Appendix B.5. We use Lemma 4.5 to assign IDs and Theorem 3.1 to simulate a powerful GNN from Loukas (2020). Despite this theoretical results, we found that AMP with constant delays works practically better and is easier to train. With constant delays, nodes receive the same messages in the same order across different runs, making for stable gradients. On the other hand, random message delays can produce different embeddings for the same graph, so it takes many representative distribution of embeddings to produce a stable gradient signal.

## 5 EXPERIMENTS

### 5.1 BEYOND 1-WL CLASSIFICATION

We experiment with AMP's ability to classify graphs that the $1-$WL test cannot distinguish. We compare on existing GNN expressiveness benchmarks for node classification (Limits1 (Garg et al., 2020), Limit2 (Garg et al., 2020), Triangles (Sato et al., 2021), and LCC (Sato et al., 2021)) and graph classification (4-cycles (Loukas, 2020), Skip-Cycles (Chen et al., 2019), Rook-Shrikande (Chen et al., 2019)). Furthermore, we test models if they can separate certain graphs from Xu et al. (2019) if we limit aggregation to `max` or `mean`. We compare AMP-RNN with several powerful GNN extensions from literature: PPGN (Maron et al., 2019), SMP (Vignac et al., 2020),[1] DropGNN (Papp et al., 2021),[2] and ESAN (Bevilacqua et al., 2022),[3] plus a simple GIN (Xu et al., 2019) for control. We needed to slightly modify all codebases. For training, we follow the setup by Papp et al. (2021). GNNs have 4 layers, for Skip-Cycles, we additionally try 9 layers and take the better result. For AMP, we allow a total of $5n$ messages, with $n$ being the size of the graph. We try both constant (AMP-C) and random message delays (AMP-R). We experiment with the simple recurrent form of AMP that uses just a linear layer for $\delta$ and $\mu$. Figure 3c shows that the starting node matters, so we execute multiple runs for AMP, one for every node with that node being the starting node. Each run computes the final embedding for the starting node. We use 16 hidden units for Limits1, Limits2, Triangles, LCC, and 4-cycles and 32 units for MAX, MEAN, and Skip-Cycles. Training uses the Adam (Kingma & Ba, 2015) optimizer with a learning rate of $0.01$.

| Dataset | GNN | PPGN | SMP | DropGNN | ESAN | AMP-C | AMP-R |
|---|---|---|---|---|---|---|---|
| Limits1 | 0.50 ±0.00 | 0.60±0.21 | 0.95±0.16 | **1.00**±**0.00** | **1.00**±**0.00** | **1.00**±**0.00** | 0.89±0.11 |
| Limits2 | 0.50±0.00 | 0.85±0.24 | **1.00**±**0.00** | **1.00**±**0.00** | **1.00**±**0.00** | **1.00**±**0.00** | **0.98**±**0.01** |
| Triangles | 0.52±0.15 | **1.00**±**0.02** | 0.97±0.11 | 0.93±0.13 | **1.00**±**0.01** | **1.00**±**0.01** | 0.88±0.15 |
| LCC | 0.38 ±0.08 | 0.80±0.26 | 0.95±0.17 | **0.99**±**0.02** | **0.96**±**0.06** | **0.96**±**0.03** | 0.80±0.04 |
| MAX | 0.05±0.00 | 0.36±0.16 | 0.74±0.24 | 0.27±0.07 | 0.05±0.00 | **1.00**±**0.00** | 0.37 ±0.10 |
| MEAN | 0.28±0.31 | 0.39±0.21 | 0.91±0.14 | 0.58±0.34 | 0.18±0.08 | **1.00**±**0.00** | 0.71±0.11 |
| 4-cycles | 0.50±0.00 | 0.80±0.25 | 0.60±0.17 | **1.00**±**0.01** | 0.50±0.00 | **1.00**±**0.00** | **0.99**±**0.02** |
| Skip-Cycles | 0.10±0.00 | 0.04±0.07 | 0.27±0.05 | 0.82±0.28 | 0.40±0.16 | **1.00**±**0.00** | 0.56±0.18 |
| Rook-Shrikande | 0.50±0.00 | 0.50±0.00 | 0.50±0.00 | 0.50±0.00 | **1.00**±**0.00** | **1.00**±**0.00** | 0.50±0.50 |

Table 1: Test set accuracy on GNN expressiveness benchmarks that require beyond $1-$WL expressiveness to solve. AMP solves all benchmarks quasi-perfect, even the challenging ones that restrict the aggregation (MAX, MEAN) or require long-range propagation (Skip-Cycles).

Table 1 shows the test graph results. AMP-C performs better than all other methods and consistently solves all datasets (close to) perfectly. Many methods perform very well on the node classification

---

[1]Code for SMP and PPGN from `https://github.com/cvignac/SMP`

[2]Code for GIN and DropGNN from `https://github.com/KarolisMart/DropGNN`

[3]Code for ESAN from `https://github.com/beabevi/ESAN`

tasks. On the graph classification tasks, all methods but AMP-C struggle. This is particularly true for the two datasets with restrictions on the aggregation. However, sometimes it is desirable not to use `sum` as aggregation function. Other aggregations, such as `max` might offer better algorithmic alignment (Xu et al., 2020; 2021). Moreover, AMP-C solves the Skip-Cycles dataset perfectly which requires long-range information propagation as well as the rook/shrikande graphs that need power beyond $3 - WL$. On the other hand, AMP-R suffers from unstable gradients and produces only mediocre results.

## 5.2 Long-range information propagation

In this section, we investigate the long-range information propagation of AMP. We experiment with a simplified version of finding shortest paths in graphs. Finding shortest paths is interesting for long-range information propagation since it requires reading the entire graph in the worst case and has been used in previous works (Tang et al., 2020; Velickovic et al., 2020; Xu et al., 2021). We simplify the shortest path setting: instead of regressing the exact distance, we classify if the shortest path is even or odd. We do this to abstract from the need to do accurate computations of distances. Neural networks struggle with precise arithmetic computations in general (Faber & Wattenhofer, 2020; Heim et al., 2020; Madsen & Johansen, 2020; Trask et al., 2018). We compare AMP with several synchronous GNNs (IterGNNs (Tang et al., 2020),[4] Universal Transformers (Dehghani et al., 2018), and NEG (Neural Execution of Graph Algorithms (Velickovic et al., 2020)). For GNNs, we mark the starting node by giving it distinct features, for AMP, we sent this node the initial message.

We train on 25 randomly created graphs with 10 trees. Graphs are based on a random spanning tree to which we add $\frac{n}{5}$ extra edges. Training runs for 1000 iterations and uses the Adam optimizer with a learning rate of 0.01. The hidden size dimension is 30. We train on graphs with size 10, then we follow previous work (Tang et al., 2020; Velickovic et al., 2020; Xu et al., 2021) to test the ability to extrapolate the solution to larger graphs. We test with graphs of sizes 10, 15, 25, 50, 100, 250, 500, 1000. Larger graphs are also more challenging since the shortest paths grow in length. We report the classification accuracy per graph size in Table 2.

We can see that all AMP versions perform better than the synchronous baseline GNNs. AMP-Iter performs especially well. We hypothesize lies in better algorithmic alignment (Xu et al., 2020; 2021) between the asynchronous architecture and the task. that this is because the asynchronous model aligns better with the given task. Algorithmic alignment has been shown to both learn with fewer samples and extrapolate better to unseen and harder instances. The better alignment in this task is that nodes in AMP only need to act when there might be relevant information. On the other hand, nodes in the synchronous GNNs need to stay ready over many rounds until the information from the starting node finally reaches them. Furthermore, the transformer-based architecture AMP-ATT seems to be misaligned with the dataset and possibly the experimental setup that trains for rather few epochs on little data (usually transformers excel on large datasets with plenty of training time).

To give some intuition why AMP works generally better on propagating information over long distances. Consider a node $v$ with degree $\delta$ that needs to receive information on the shortest way from a node that has distance $d$ to $v$. In a GNN, $v$ needs to receive $d\delta$ many messages to receive the one message. In AMP, $v$ will have received the message after hearing from every neighbor at most once, which are $\delta$ many messages in the worst case (independent of $d$, which is often dependent on the graph size). We discuss this idea in more detail in Appendix C and relate to the problems of Underreaching, Oversmoothing, and Oversquashing.

## 5.3 Graph Classification

Last, we train AMP on several graph classification benchmarks. Our AMP implementation runs on a single CPU since the sequential nature limits GPU potential and python's Global-Interpreter-Lock[5] prevents multithreading. This makes training larger datasets prohibitively slow for now (for example, PROTEINS takes around $1 - 2$ days to train). Therefore we limit the comparison to the smaller datasets MUTAG, PTC, PROTEINS, IMDB-B, IMDB-M (Yanardag & Vishwanathan, 2015). Further, we do a single pass in each training epoch instead of 50 batches of 50 graphs. We run AMP-RNN

---

[4]Code from IterGNN from `https://github.com/haotang1995/IterGNN`
[5]`https://wiki.python.org/moin/GlobalInterpreterLock`

| Model | 10 | 25 | 50 | 100 | 250 | 500 | 1000 | 2500 |
|---|---|---|---|---|---|---|---|---|
| NEG | 0.68±0.16 | 0.56±0.08 | 0.52±0.03 | 0.49±0.02 | 0.49±0.01 | 0.49±0.01 | 0.50±0.01 | 0.49±0.00 |
| Universal | 0.91±0.05 | 0.73±0.05 | 0.62±0.03 | 0.57±0.02 | 0.53±0.01 | 0.51±0.00 | 0.50±0.00 | 0.50±0.00 |
| IterGNN | 0.85±0.11 | 0.68±0.06 | 0.58±0.03 | 0.55±0.02 | 0.52±0.00 | 0.51±0.00 | 0.50±0.00 | 0.50±0.00 |
| AMP-RNN | 0.81±0.16 | 0.73±0.17 | 0.65±0.16 | 0.61±0.14 | 0.54±0.10 | 0.54±0.08 | 0.54±0.07 | 0.53±0.04 |
| AMP-GRU | **0.98±0.01** | 0.90±0.05 | 0.83±0.07 | 0.75±0.10 | 0.63±0.13 | 0.57±0.14 | 0.54±0.10 | 0.52±0.08 |
| AMP-LSTM | **0.98±0.02** | 0.89±0.10 | 0.8±0.10 | 0.74±0.12 | 0.66±0.12 | 0.63±0.11 | 0.61±0.07 | 0.58±0.06 |
| AMP-ATT | 0.48±0.01 | 0.50±0.01 | 0.49±0.01 | 0.50±0.00 | 0.50±0.00 | 0.49±0.00 | 0.49±0.00 | 0.49±0.00 |
| AMP-ACT | **1.00±0.00** | **0.99±0.01** | **0.98±0.01** | **0.97±0.02** | **0.96±0.03** | **0.95±0.04** | **0.95±0.05** | **0.94±0.05** |
| AMP-Iter | **1.00±0.00** | **0.99±0.00** | **0.98±0.03** | **0.97±0.04** | **0.96±0.05** | **0.96±0.06** | **0.95±0.07** | **0.94±0.08** |

Table 2: Accuracy for predicting the parity of shortest paths to a starting node. The table head contains the number of nodes in the test graph while training is always on 10 nodes. AMP-Iter learns to extrapolate almost perfectly. Also other AMP variants extrapolate better than the GNN baselines.

which does a short run from every node. We do a small grid search over the number of messages (15 or 25) per run, the size of the message embeddings (10 or half the node embedding size), and whether AMP-RNN uses skip connections to previous states. The remaining setup is taken from Xu et al. (2019): We run on 10 different splits and report the accuracy of the best performing epoch. We use hidden node states of 64 for social datasets and 16 or 32 for biological datasets. We further compute results for GraphSAGE (Hamilton et al., 2017), GCN (Kipf & Welling, 2017), and GAT (Veličković et al., 2018) and take results for more GNNs. Table 3 shows all results. Even with little investigation into AMP architectures and hyperparameters, AMP achieves comparable results to existing GNNs but is not quite state-of-the-art. We believe that further improvements in AMP will reach a competitive performance.

| Model | MUTAG | PTC | PROTEINS | IMDB-B | IMDB-M |
|---|---|---|---|---|---|
| PatchySan (Niepert et al., 2016) | **92.6 ±4.2** | 62.3 ±5.7 | 75.9 ±2.8 | 71.0 ±2.2 | 45.2 ±2.8 |
| DGCNN (Zhang et al., 2018) | 85.8 ±1.7 | 58.6 ±2.5 | 75.5 ±0.9 | 70.0 ±0.9 | 47.8 ±0.9 |
| GraphSAGE (Hamilton et al., 2017)* | 90.4±7.8 | 63.7±9.7 | 75.6±5.5 | 76.0±3.3 | 51.9±4.9 |
| GCN (Kipf & Welling, 2017) | 88.9 ±7.6 | **79.1 ±11.4** | 76.9 ±4.8 | **83.4 ±4.9** | **57.5±2.6** |
| GAT (Veličković et al., 2018) | 85.1 ±9.3 | 64.5±7.0 | 75.4±3.8 | 74.9±3.8 | 52.0±3.0 |
| GIN (Xu et al., 2019) | 89.4 ±5.6 | 66.6 ±6.9 | 76.2 ±2.6 | 75.1 ±5.1 | 52.3 ±2.8 |
| 1-2-3 GNN (Morris et al., 2019) | 86.1 | 60.9 | 75.5 | 74.2 | 49.5 |
| DropGNN (Papp et al., 2021) | 90.4 ±7.0 | 66.0±9.8 | 76.3 ±6.1 | 75.7 ±4.2 | 51.4 ±2.8 |
| PPGN (Maron et al., 2019)* | 90.6 ±8.7 | 66.2 ±6.5 | 77.2 ±4.7 | 73 ±5.8 | 50.5 ±3.6 |
| ESAN (Bevilacqua et al., 2022)* | 92.0±5.0 | 69.2±6.5 | **77.3 ±3.8** | 77.1±2.6 | 53.7±2.1 |
| AMP-RNN (ours) | 90.4±4.1 | 63.7 ±9.1 | 76.7±7.1 | 74.6±3.6 | 52.1±3.6 |

Table 3: Graph classification accuracy (%). First block are 1− WL GNNs, second block are beyond 1−WL GNNs. AMP produces results that are comparable to both GNN variants. *We report the result achieved by the best model version.

## 6 CONCLUSION

We presented a new framework, AMP, for learning neural networks on graphs. AMP can treat every message individually, and as such (in principle) no information is lost in aggregation. This benefits AMP to tackle problems such as underreaching or oversquashing and makes AMP not limited to the 1−WL test. We theoretically investigate its expressiveness and empirically evaluate its ability to distinguish hard graphs, propagate information over long distances, and classify common benchmarks. AMP might also be more interpretable if we investigate when nodes react or ignore a message. This makes us believe that AMP is a promising paradigm that warrants further exploration. GNNs have improved considerably over the last years and we believe AMP will similarly. However, before we can apply AMP to most real-world problems, we need to improve tooling to allow parallelism to reduce the problematic runtimes for larger graphs.

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

# A    MORE DETAILS ON AMP

## A.1    ANOTHER PERSPECTIVE ON AMP

Rather than looking at all nodes at the same time, let us view the AMP execution from the perspective of a single node (Figure 4). This node receives a sequence of messages and transitions through a series of states and produces new messages. In cases learnt through $\rho$, the node skips the input (for example for message $m_{i+1}$). The behavior mimics that of sequence-to-sequence models that have seen great success in Natural Language Processing (Sutskever et al., 2014). Compared to NLP sequence-to-sequence models, the AMP must overcome two additional challenges.

**Unknown input sequences.** In most sequence-to-sequence language models, we want to transform a known input sequence into an output sequence. In NLP, input sequences are often sentences and the sequence elements are words, for example, when we want to translate sentences. In such cases, the input sequence is entirely known at the start. In the context of Figure 4, a language model can learn to optimize the sequence of $h_i$ while the sequence of $m_i$ is constant. In AMP, the messages are not constant but need to be jointly optimized with the state update function. This makes the gradient signals noisier and the training process more difficult.

**Partial information accessible.** In natural language processing, the input sequence contains all the needed information to produce the target prediction. Consider again the translation setting, where sentences are inputs and words are sequence elements. Every word has access to the entire input sequence and thus all the information it needs. However, most nodes in graphs do not have access to all information they need. Figure 5 illustrates the problem with a simple example. The node $v$ on the right needs to know if there is a red node in the graph. There is one such node, $s$. However, in between those two nodes are nodes $t$ and $u$. For both $t$ and $u$, the red node $s$ is not important since $t$ and $u$ are only interested in the number of blue nodes. Therefore, if either $t$ or $u$ decide to not forward information about the red node, the information that $s$ exists never reaches $v$ and $v$ cannot classify correctly. The takeaway from this example is that nodes not only need to understand what messages are important to them and how to use these messages, but nodes also need to identify the messages that are important to other nodes and propagate these messages.

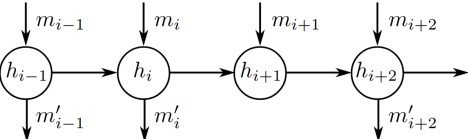

Figure 4: AMP dynamics for one node in isolation. The node receives a sequence of messages which it can use to update its state and emit a sequence of messages.

Figure 5: Node $v$ needs a message from $s$ but needs the unrelated $t$ and $u$ to forward the message.

## A.2    CONCRETE MODELS IN THE AMP FRAMEWORK.

The analogy to recurrent architectures lens ideas for several implementations of the AMP framework. Probably among the simplest is the equivalent of a recurrent neural network: we skip $\rho$ and use linear layers for $\delta$ and $\mu$. Other feedforward layers such as multilayer perceptrons or other neural architectures are equally possible. Upon receiving messages $m_1, m_2 \ldots m_i$, we compute the hidden states $z_i = \delta(z_{i-1}, m_i)$ with $z_0$ being the initial features and $m_i = \mu(z_i, m_i)$.

We can also instantiate the AMP framework with other recurrent architectures. For example, we can create a GRU cell to model $\delta$ and input received messages as they come in over time (Cho et al., 2014). The gating of the GRU cell also implicitly models the $\rho$ function since it can learn to ignore certain messages. Hidden states are computed as $z_i = \text{GRU}(z_{i-1}, m_i)$. We make the message computation a linear transformation of $z_i$. Analogously, we can create a version of AMP using an LSTM cell (Hochreiter & Schmidhuber, 1997).

Another version of AMP could use the attention mechanism of transformers (Vaswani et al., 2017) to compute node states based on the history of messages received so far. When receiving message

$m_i$, we compute:

$$z_i = \delta(z_{i-1}, \sum_{k=1..i} \alpha(z_{i-1}, m_k))$$

The function $\alpha$ is a parametrized dot-product attention function that computes an attention score for every previous message and normalizes them to sum up to 1. We use a linear readout for the new message. This model can also implicitly model $\rho$ by assigning very small attention values to unimportant messages.

Let us now look at two example architectures that explicitly model $\rho$. The first idea is based on the idea of Adaptive Computation Time by Graves (2016). Upon every received message, the receiver computes a termination probability that this message should be its last. When the sum over all termination probabilities is at least $1 - \epsilon$, the node stops being interested in further messages and computes a final embedding based on a termination-probability-weighted average. Upon receiving message $m$ we compute:

$$\rho_{i+1} = \min(1 - \sum_{k\in 1..i} \rho_k, \rho(\delta_i, m))$$

Where we have $\delta_0$ as the initial node state. We compute the final representation after $n$ steps as:

$$\delta = \sum_{k\in 1..n} \rho_k \cdot \delta_k$$

The second idea follows Tang et al. (2020) and also uses termination probabilities. However, probabilities are combined multiplicatively instead of additively:

$$\rho_{i+1} = \prod_{k=1..i} (1 - \rho_i) \cdot \rho(\delta_i, m)$$

Again, we can compute the final representation as a termination-probability weighted average of all intermediate values. Both ideas for $\rho$ make no assumption about $\delta$ or $\mu$.

# B PROOFS

## B.1 PROOF FOR THEOREM 3.1

The underlying proof idea is the $\alpha$ synchronizer from asynchronous distributed computing by Awerbuch (1985). In this synchronizer, nodes are only allowed to send messages when they are "safe". A node is safe when it received an acknowledgment from every neighbor for its previous message. The authors show that the $\alpha$ synchronizer allows asynchronous models to mimic distributed models such as LOCAL (Peleg, 2000).

We will give a constructive proof of $\mathcal{A}_{\mathcal{G}}$. Let $\mathcal{G}$ be a message passing GNN with MESSAGE and UPDATE functions. Let us further assume that every layer in $\mathcal{G}$ has the same dimension $d$ (for simplicity, we handle different embedding sizes later) and messages have size $m$. Let $L$ be the total number of layers in $\mathcal{G}$ and $D$ be the degree for a node[6].

We construct $\mathcal{A}_{\mathcal{G}}$ as follows: $\mathcal{A}_{\mathcal{G}}$ will have states of size $2d + 3$ that we represent as tuples $(s, a, w, u, i, l)$ with the following semantics: $l$: The number of the last layer of $\mathcal{G}$ that we have simulated, $u$ The number of "unacked" neighbors of a node, $w$: The number of messages that a node is waiting on, $i$, whether the node acted at least once (only true until the first update), $s$: The actual state of the node. This will always correspond to the representation in $\mathcal{G}$ at the layer mapped by $l$ of the node, and $a$: An accumulator to collect the messages of the node's neighbors. We initialize s to $(\boldsymbol{x}, 0, D-1, D, 0, L)$ with $\boldsymbol{x} \in \boldsymbol{X}$ in $\mathcal{G}$.

Furthermore, $\mathcal{A}$ sends messages of size $m + 4$ which we represent as tuples $(m, ack, pulse, origin, noop)$. The last four positions one-hot encode which kind of message we send. For $\mathcal{A}_{\mathcal{G}}$, we choose $\mu = $ MESSAGE and set $m = \mu(s)$. We set $\rho$ to ignore messages of type "noop" or when simulated all $L$ many layers of $\mathcal{G}$. We provide $\delta$ and which message type to send in Table 4. Image the function in the table as a cascade of conditional assignments which we evaluate from top to bottom and stop on the first matching one. We will show that whenever a node $v$ emits a "pulse" message, its current state $s$ maps to the message passing embedding $z_v^{L-l}$. Nodes will emit such "pulse" messages when (i) they received a "pulse" message from every neighbor (ii) they received an "ack" message from every neighbor (when they are safe).

| Condition | s' | a' | w' | u' | i' | l' | message type |
|---|---|---|---|---|---|---|---|
| noop=1 | $\perp$ | $\perp$ | $\perp$ | $\perp$ | $\perp$ | $\perp$ | $\perp$ |
| l=0 | $\perp$ | $\perp$ | $\perp$ | $\perp$ | $\perp$ | $\perp$ | $\perp$ |
| origin=1 | s | a | w | D | 0 | l | pulse |
| i=1 | s | a+m | w-1 | u | 0 | l | pulse |
| pulse=1 $\wedge$ w=0 | UPDATE([s,(a+m)]) | 0 | D-1 | u-1 | 0 | l-1 | ack |
| pulse=1 | s | a+m | w-1 | u | 0 | l | noop |
| ack=1 $\wedge$ u=0 | s | a | w | D | 0 | l | pulse |
| ack=1 | s | a | w | u-1 | 0 | l | noop |

Table 4: Update function $\delta$ and message type to send (part of $\mu$ used by every node in AMP to simulate a synchronous GNN.

**Lemma B.1.** *When a node executes the $i = 1$ condition it received a pulse message.*

*Proof.* There are four possible messages origin, pulse, noop, and ack. Every noop message is ignored, and the origin message is handled differently and only sent once. Let us assume that a node $v$ receives an ack message before a pulse message. This would require that one of $v$'s neighbor $u$ received one message from every neighbor once, includeing $v$. However $v$ did not act yet since it just received its first message so $v$ cannot have sent such a message. $\square$

**Lemma B.2.** *When node $v$ emits its $i-$ pulse message, it must have already received $(i-1) \cdot D+1$ pulse messages[7].*

---

[6]We take the node degree as a given which simplifies the proof but does not leak information to $\mathcal{A}_{\mathcal{G}}$. Nodes in AMP can their degree by flooding the network once and storing the count in an additional state entry that is never modified again.

[7]There is one exception to this lemma, which is the node receiving the "origin" in place for one pulse message.

*Proof.* We prove by induction. The lemma is true after initialization when all nodes are about to send their 1 pulse messages. Nodes send a `pulse` message when they receive one themselves, thus, after 1 total.

Suppose the lemma is true for $i$ and node $v$ has received $(i - 1) \cdot D + 1$ and just sent its $i-$ `pulse` message. For $i > 1$ this is only possible by the seventh condition in Table 4, which also sets $u = D$. Before we can send another `pulse` via this condition $u$ must be decremented to 0. This requires an "ack" message from every neighbor and also executing the fifth condition, which requires $w = 0$. Decrementing $w$ to zero is only possible by receiving $D - 1$ `pulse` message in the sixth condition and then one more message in the fifth condition. At this point, $v$ has received $(i - 1) \cdot D + 1 + D - 1 + 1 = i \cdot D + 1$ pulse messages. $\qquad\square$

**Corollary B.3.** *If nodes require $D$ `pulse` messages between their own pulses, it receives one pulse each from every neighbor.*

**Lemma B.4.** *When a node $v$ emits its $i-th$ `pulse` message, it's state $s$ equals the synchronous GNN representation $z_v^{L-i-1}$.*

*Proof.* We proof by induction. Because of the way nodes are initialized, the moment they receive their first message (condition 2 or 3 nodes emit their first `pulse`). At this point, the node state equal the initial features aka $z_v^0$.

Assume the lemma holds for node $v$ just send it's $i-$ pulse message. We know from Lemma B.2, that $v$ now needs to receive $D$ pulse messages before it can send pulse $i + 1$. One pulse will come from each neighbor following B.3. According to the induction hypothesis, node $v$ receives $z_w^{l-i-1}$ from every neighbor $w$ in their $i$th pulse message. Upon receiving the last of these messages (fifth condition), $v$ computes locally UPDATE($z_v^{l-i-1}, \sum_w z_w^{l-i-1} = z_v^{l-i}$). This state is unchanged until $v$ emits its $i + 1$st pulse. $\qquad\square$

We can gather all representations that $\mathcal{G}$ can compute by listening to each node and recording its state when it emits a pulse message.

**Disconnected graphs**    The proof assumes a connected graph but we can extend it to arbitrary graphs. We can either (i) send one origin message to each connected component or (ii) introduce a master node that connects to every node (thus connecting the graph) and does nothing but forward pulse messages while ignoring all other messages. This master node does not count into a node's degree.

**Differing layer sizes**    Let us relax the assumption that we have a uniform width $d$ over all layers and instead different widths $d_1, d_2, \ldots d_L$. We can still use the above construction, but the AMP model will have a state space consisting of $s_0, a_0, s_1, a_1, \ldots s_L, a_L, w, u, l$. Furthermore, the update in the sixth condition does not overwrite the state but writes in the next layer's state. Similarly, we can support different message sizes and different message functions per layer.

**Skip connections**    The construction for different layer sizes computes and stores the embeddings for every layer. If we modify $\delta$ to not overwrite correctly computed values (for example by only updating state vectors with a matching value of $l$), we can also simulate skip connections.

**Other aggregation functions**    Other associative aggregation functions work out of the box with the above construction. But we can also other aggregations work, for example, mean aggregation. For mean aggregation, we do not only keep track of the sum but also the count of messages in the aggregator. Then we compute the UPDATE function on the count divided by the sum. Similarly, we can extend, for example, to the GCN (Kipf & Welling, 2017) model.

**Learning the function in Table 4**    According to the universal approximation theorem (Royden & Fitzpatrick, 1988), using a sufficiently wide $2-$ layer neural network can approximate any function. Therefore, we can choose $\delta$ and $\mu$ to be such networks to model the function shown by Table 4.

## B.2 Proof for Lemma 4.2

*Proof.* Let the computation start from node $v$ which is part of the cycle and every node executes the following protocol. If a node receives a message COUNT$-i$ and it never received a message before, it stores $i$ and messages COUNT$-(i + 1)$. If the same node then receives a COUNT$-j$ message with $j > i$ the node ignores the message. Node $v$ starts the computation by sending COUNT$-0$. The nodes on both paths in the cycle now start storing numbers mapping to a BFS. Eventually, the two BFS branches meet. This happens when one node $w$ that first received a COUNT$-i$ message receives a COUNT$-j$ message with $i \geq j$. Then $w$ knows that the circle is closed and sends a FOUND$-(j + i)$ that every node forwards once. Thus, all nodes become aware of the cycle and its length. $\qquad\square$

## B.3 Proof for Lemma 4.3

*Proof.* $v$ and its neighbors will iteratively find out that they are in $2, 3, \ldots k + 1$ cliques. The starting node $v$ will coordinate the other nodes. Initially, every node stores that they are in a $1-$clique. To find a clique of size $j$, node $v$ sends a CLIQUE$-j$ message, which every neighbor of $v$ forwards once. If neighbors of $v$ receive $j - 1$ such messages and they are in a $j - 1$ clique according to their state, they send a CLIQUE$-j-$ACK message to all neighbors (including $v$) and update their state to be in a $j-$clique. If $v$ receives $k$ many CLIQUE$-j-$ACK messages, it sends out a CLIQUE$-(j+1)$ message. Upon receiving $k$ many CLIQUE$-(k + 1)-$ACK messages, $v$ knows there exists a $(k + 1)-$clique and can propagate this information to its neighbors. $\qquad\square$

## B.4 Proof for Lemma 4.5

We show that AMP with random delays can create unique identifiers for every node in a star graph with central node $v$ and $k$ outer nodes. The problem we need to solve is bringing order in the equal neighbors of $v$, which we do by exploiting the random delays: We connect the neighbors of $v$[8]. When $v$ "offers" an ID, nodes without an ID will reply to try to claim it. Some nodes will receive this reply before they receive the message from $v$. These nodes will surrender this ID. After some attempts to offer the ID, all but one node will surrender. Then $v$ gives the ID to the one non-surrendering node (implicitly by starting to offer the next ID). Then the protocol restarts for the next ID (minus the nodes that are done and have an ID).

We prove the Lemma constructively by transforming this idea into a network. We have four states, offering (which is only used by $v$), claiming and surrendering (which are used by the outer nodes), and done used by all nodes that no longer need to participate. We have five message types offer, confirm, claim, surrender, origin, and noop.

Nodes have states that are tuples (state, try, ID, c, w, x): state is one of above states, try identifies different ID assignment attempts from each other, and ID is the ID of the node. The next three entries are only used by central node $v$: c a binary feature if the ID was already claimed in the current attempt, w contains the number of neighbors in the current attempt that did not reply yet, x contains the number of neighbors that do not have an ID yet. Initially, every node starts in state (surrendering, -1, -1, 0, 0, 0).

Messages are tuples (type, CID, attempt): type is one of above messages types, CID is a candidate ID that $v$ tries to currently assign, attempt is the current attempt to assign an ID.

Table 5 shows the node update and message functions in AMP for the center node, Table 6 for the outer nodes. Practically there is of course only one function and nodes can judge by their state if they are the center node or node. For readability, we split them here. If the reaction function $\rho$ decides to ignore a message, the tables show a row of blanks ($\perp$).

*Proof.* Let us now prove that his algorithm assigns valid identifiers, namely (i) that no two nodes receive the same identifier and (ii) every node receives an identifier.

(i) The center node takes ID $0$ for itself and only proposes IDs of $1$ and above to its neighbors. If a neighbor does not change its ID, it stays with $-1$. Thus ID $0$ is unique. Let us suppose that two

---

[8]These edges are not necessary since $v$ could propagate messages between outer nodes, but they make the idea more concise and the algorithm faster

nodes $w_1, w_2$ received the same ID. Since IDs are increasing $w_1$ and $w_2$ must receive the ID at the same time. Since the `confirm` message is only sent when there was one node left, $w_1$ and $w_2$ must have received the ID via the second to last condition in Table 6a. This is only possible if they were in state `claiming`, which means that they both send a `claim` message. However, in such cases, $v$ starts another try with the same ID as before. This contradicts the requirement that ID$\neq$CID.

(ii) The center node receives ID $0$. We have to show all other nodes also receive an ID. Node $v$ counts internally how many IDs it could offer successfully and only stops when this counter reaches $0$. The counter is initialized with $v$ degree so $v$ will try to hand out enough IDs. Let us suppose that $w$ did not receive an ID. Unless $w$ will always try to claim an ID unless it is already surrendering this ID. Therefore $w$ surrenders for all IDs, even the last one. However, this is not possible since there is only $w$ left and no node could have sent a `claim` message to make $w$ surrender. $\square$

| Condition | state | try | ID | c | w | x | type | CID | attempt |
|---|---|---|---|---|---|---|---|---|---|
| noop | $\perp$ | $\perp$ | $\perp$ | $\perp$ | $\perp$ | $\perp$ | $\perp$ | $\perp$ | $\perp$ |
| done | $\perp$ | $\perp$ | $\perp$ | $\perp$ | $\perp$ | $\perp$ | $\perp$ | $\perp$ | $\perp$ |
| try $\neq$ attempt | $\perp$ | $\perp$ | $\perp$ | $\perp$ | $\perp$ | $\perp$ | $\perp$ | $\perp$ | $\perp$ |
| origin | offering | 0 | 0 | 0 | D-1 | D-1 | offer | 1 | 0 |
| claim $\wedge$ c = 0 | offering | try | 0 | 1 | w-1 | x | noop | $\perp$ | $\perp$ |
| claim $\wedge$ c > 0 | offering | try+1 | 0 | 0 | x-1 | x | offer | CID | try+1 |
| surrender | offering | try | 0 | c | w-1 | x | noop | $\perp$ | $\perp$ |
| w = 0 $\wedge$ x > 0 | offering | try+1 | 0 | 0 | x-2 | x-1 | offer | CID+1 | try+1 |
| x = 0 | done | 0 | 0 | 0 | 0 | 0 | confirm | $\perp$ | $\perp$ |
| (a) | | | | | | | (b) | | |

Table 5: Node update function $\delta$ (a) and message function $\mu$ (b) for the central node $v$ in AMP to assign unique IDs in a star graph.

| Condition | state | try | ID | c | w | x | type | CID | attempt |
|---|---|---|---|---|---|---|---|---|---|
| noop | $\perp$ | $\perp$ | $\perp$ | $\perp$ | $\perp$ | $\perp$ | $\perp$ | $\perp$ | $\perp$ |
| done | $\perp$ | $\perp$ | $\perp$ | $\perp$ | $\perp$ | $\perp$ | $\perp$ | $\perp$ | $\perp$ |
| surrendering $\wedge$ offer $\wedge$ CID=ID | surrendering | attempt | ID | $\perp$ | $\perp$ | $\perp$ | surrender | CID | attempt |
| offer $\wedge$ attempt > try | claiming | attempt | CID | $\perp$ | $\perp$ | $\perp$ | claim | CID | attempt |
| claiming $\wedge$ attempt > try | surrendering | attempt | CID | $\perp$ | $\perp$ | $\perp$ | surrender | CID | attempt |
| claiming $\wedge$ offer $\wedge$ ID $\neq$ CID | done | 0 | ID | $\perp$ | $\perp$ | $\perp$ | noop | $\perp$ | $\perp$ |
| claiming $\wedge$ confirm | done | 0 | ID | $\perp$ | $\perp$ | $\perp$ | noop | $\perp$ | $\perp$ |
| (a) | | | | | | | (b) | | |

Table 6: Node update function $\delta$ (a) and message function $\mu$ (b) for the outer nodes in AMP to assign unique IDs in a star graph.

### B.5 PROOF FOR THEOREM 4.7

*Proof.* We are reducing Theorem 4.7 to Corollary 3.1 from Loukas (2020). The authors show that if we have (i) unique identifiers for every node in a graph and (ii) we have a message passing GNN that has no bound in width and is at least as deep as the graph's diameter $\delta_G$, this GNN is Turing universal and can compute any function on this graph (including graph isomorphism).

Let $G_1, G_2$ be two connected graphs for which we want to test graph isomorphism. We create the graph $G$ by adding a unique node $u$[9] that we connect to a random node from $G_1$ and $G_2$ each. Further, let $\mathcal{G}$ be a GNN of unbounded width that has at least $\delta_G$ many layers. If the GNN has access to IDs, this GNN is Turing universal and can compute any computable function. Therefore, this GNN can compute if $G_1$ and $G_2$ are isomorphic. Let us consider an AMP model with unbounded width, which can compute graph isomorphism on $G$ as follows: First, we employ Lemma 4.5 to assign every node

---

[9]Without loss of generality, we can also start the ID assignment in AMP at node $u$, this node would always have ID $0$.

a unique identifier. Then we leverage Theorem 3.1 and simulate $G$ with the identifiers for $\delta_G$ many rounds. Whatever representations $\mathcal{G}$ finds to be able to compute graph isomorphism between $G_1$ and $G_2$, AMP can compute them as well through simulation. Thus AMP can also compute graph isomorphism on any graph $G$.

Finally, let us tackle the unbounded width requirement. We are not interested in any algorithm on $G$, but only in determining graph isomorphism. We can estimate the required width for this problem as follows: One example algorithm to compute graph algorithms is to use the $\delta_G$ layers to build a complete representation of $G$ in every node. If we use adjacency matrices this requires width $O(n^2)$. Now we enumerate all possible assignments $\pi$ from nodes in $G_1$ to $G_2$ — of which there are $O(\frac{n}{2}!)$ many. For each candidate we write the permuted adjacency matrix, requiring $O(\frac{n}{2}!)$ space. If any permuted adjacency matrix equals that of $G_1$, the graphs are isomorphic, otherwise they are not. We can find out, by checking if their subtraction equals $0$ which requires constant width for each of the $O(\frac{n}{2}!)$ candidates. We can conclude that the GNN can solve graph isomorphism on $G$ with $O(\frac{n}{2}!)$ width. This is the same bound for AMP that needs only a constant factor in state and message size to simulate a GNN. □

## C    Analysis of Results in Section 5.2

### C.1    Underreaching

To better understand the improvements of AMP, we analyze the exposure of all models to underreaching, oversmoothing, and oversquashing. To estimate underreaching, we break down the accuracy by distance from the starting node. If accuracy decreases with increasing distance it suggests exposure to underreaching. Practically, since the label flips in every step we merge an odd and an even pair into one bucket. For example, the scores for distances 1 and 2 are combined. Figure 6 shows the results broken down by distance. We can see that almost no architecture except AMP-Iter and AMP-ACT can extrapolate to much larger distances than the training set. This contributes largely to the decrease in accuracy since the larger graphs become, the more nodes have large distances to the starting node.

To gain some mathematical intuition into the results, let us understand the task a bit better: A node $v$ with degree $\delta$ and distance $d$ to the starting node, only neds to know the parity of its predecessor $u$ on the shortest path from $s$ to $v$. If $u$ has an even parity, $v$ has odd parity and vice versa. Therefore, $v$ needs exactly one message (assuming shortest paths are unique, otherwise there might be multiple useful messages). However, $v$ will receive at least $d \cdot \delta$ many messages in a synchronous GNN before it can classify. Here, $\delta$ grows with increasing graph size which means the useful message is lost in more and more unimportant messages. On the other hand, nodes in AMP can decide after they received a message for each neighbor once, in other words after $\delta$. In the simpler AMP-RNN, the same neighbors might send multiple messages which they do not need to do in AMP-Iter (nodes ignore unimportant messages and terminate after the message from $u$). This might explain the performance difference between those two architectures.

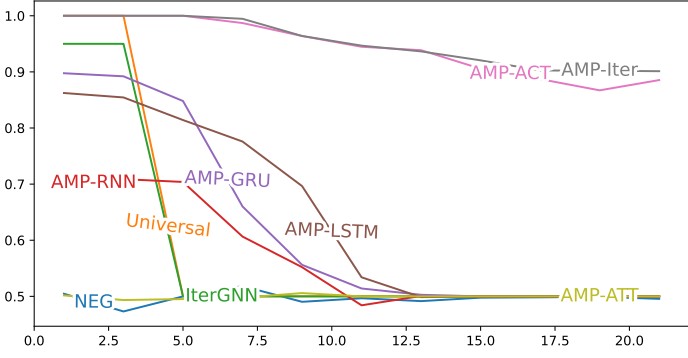

Figure 6: Caption

| Distance | NEG | Universal | IterGNN | AMP-RNN | AMP-GRU | AMP-LSTM | AMP-ATT | AMP-ACT | AMP-Iter |
|---|---|---|---|---|---|---|---|---|---|
| 0-1 | 0.50±0.04 | 1.0±0.0 | 0.95±0.07 | 0.69±0.09 | 0.89±0.07 | 0.86±0.08 | 0.50±0.00 | 1.0±0.0 | 1.0±0.0 |
| 2-3 | 0.47±0.05 | 1.0±0.0 | 0.95±0.07 | 0.70±0.10 | 0.89±0.07 | 0.85±0.09 | 0.49±0.00 | 1.0±0.0 | 1.0±0.0 |
| 4-5 | 0.49±0.01 | 0.5±0.0 | 0.5±0.0 | 0.70±0.10 | 0.84±0.08 | 0.81±0.11 | 0.49±0.01 | 1.0±0.0 | 1.0±0.0 |
| 6-7 | 0.51±0.03 | 0.5±0.0 | 0.5±0.0 | 0.60±0.10 | 0.66±0.17 | 0.77±0.11 | 0.49±0.00 | 0.98±0.01 | 0.99±0.00 |
| 8-9 | 0.49±0.01 | 0.5±0.0 | 0.5±0.0 | 0.55±0.11 | 0.55±0.11 | 0.69±0.10 | 0.50±0.01 | 0.96±0.02 | 0.96±0.03 |
| 10-11 | 0.49±0.01 | 0.5±0.0 | 0.5±0.0 | 0.48±0.03 | 0.51±0.02 | 0.53±0.02 | 0.5±0.0 | 0.94±0.02 | 0.94±0.04 |
| 12-13 | 0.49±0.01 | 0.5±0.0 | 0.5±0.0 | 0.50±0.00 | 0.50±0.00 | 0.49±0.00 | 0.5±0.0 | 0.93±0.03 | 0.93±0.04 |
| 14-15 | 0.49±0.00 | 0.5±0.0 | 0.5±0.0 | 0.5±0.0 | 0.5±0.0 | 0.5±0.0 | 0.5±0.0 | 0.90±0.05 | 0.92±0.06 |
| 16-17 | 0.49±0.00 | 0.5±0.0 | 0.5±0.0 | 0.5±0.0 | 0.5±0.0 | 0.5±0.0 | 0.5±0.0 | 0.88±0.06 | 0.90±0.07 |
| 18-19 | 0.49±0.00 | 0.5±0.0 | 0.5±0.0 | 0.5±0.0 | 0.5±0.0 | 0.5±0.0 | 0.5±0.0 | 0.86±0.08 | 0.90±0.07 |
| 20-21 | 0.49±0.00 | 0.5±0.0 | 0.5±0.0 | 0.5±0.0 | 0.5±0.0 | 0.5±0.0 | 0.5±0.0 | 0.88±0.08 | 0.90±0.07 |

Table 7: Investigation of the underreaching effect. Table columns show the accuracy per metric for nodes with increasing distance to the starting node. If a loss in accuracy occurs, the model has trouble transmitting the information to larger distances. AMP-based approaches are better equipped to transmit information to long distances, in particular AMP-ACT and AMP-Iter.

## C.2 OVERSMOOTHING

To estimate oversmoothing, we restrict the accuracy measurement towards nodes whose distance to the starting node is in the training set. If the accuracy decreases, the presence of far away nodes in larger graphs impacts close-by nodes, which indicates oversmoothing. Figure 7 shows the results of this analysis. In principle, all models combat oversmoothing well except AMP-ATT and NEG and to a lesser extent AMP-RNN.

To gain some intuition into the oversmoothing results let us look at messages sent after a node $v$ received the message it needed to classify. In GNNs the node can decide to participate in $L - \delta$ more rounds (where $L$ is an upper limit on the number of layers). However, $v$ does not learn any new information in these layers, they can only disturb the signal already gathered. In IterGNN and Universal the node $v$ can learn to stop listening thus avoiding this effect. Similarly, AMP-ACT and AMP-Iter stop avoid oversmoothing.

AMP-GRU and AMP-LSTM can also learn to ignore later messages but they need to learn for every possible message to ignore the message via gating. This is harder than AMP-ACT or AMP-Iter, which use up a budget so *any* later message is ignored. Last, AMP-RNN has no such mechanism so we see it slowly degrading.

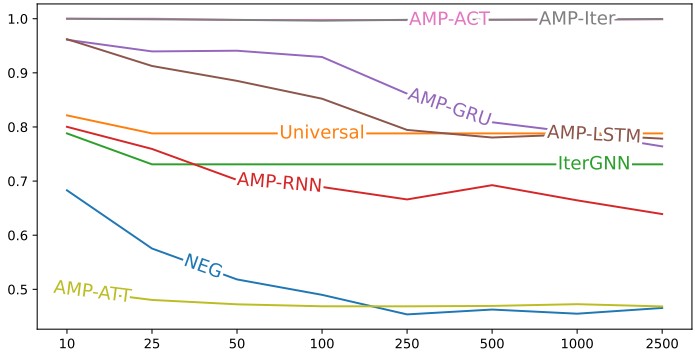

Figure 7: Caption

| Model | 10 | 25 | 50 | 100 | 250 | 500 | 1000 | 2500 |
|---|---|---|---|---|---|---|---|---|
| NEG | 0.68±0.17 | 0.57±0.09 | 0.51±0.05 | 0.48±0.11 | 0.45±0.08 | 0.46±0.08 | 0.45±0.10 | 0.46±0.08 |
| Universal | 0.82±0.03 | 0.78±0.06 | 0.78±0.06 | 0.78±0.06 | 0.78±0.06 | 0.78±0.06 | 0.78±0.06 | 0.78±0.06 |
| IterGNN | 0.78±0.10 | 0.73±0.06 | 0.73±0.06 | 0.73±0.06 | 0.73±0.06 | 0.73±0.06 | 0.73±0.06 | 0.73±0.06 |
| AMP-RNN | 0.80±0.17 | 0.75±0.19 | 0.70±0.24 | 0.68±0.24 | 0.66±0.25 | 0.69±0.22 | 0.66±0.20 | 0.63±0.15 |
| AMP-GRU | 0.96±0.05 | 0.93±0.05 | 0.94±0.07 | 0.92±0.09 | 0.86±0.23 | 0.80±0.28 | 0.78±0.26 | 0.76±0.28 |
| AMP-LSTM | 0.96±0.06 | 0.91±0.11 | 0.88±0.15 | 0.85±0.18 | 0.79±0.24 | 0.78±0.28 | 0.78±0.28 | 0.77±0.28 |
| AMP-ATT | 0.50±0.03 | 0.48±0.04 | 0.47±0.03 | 0.46±0.03 | 0.46±0.03 | 0.46±0.03 | 0.47±0.03 | 0.46±0.03 |
| AMP-ACT | 1.0±0.0 | 1.0±0.0 | 0.99±0.00 | 0.99±0.00 | 0.99±0.00 | 0.99±0.00 | 0.99±0.00 | 0.99±0.00 |
| AMP-Iter | 1.0±0.0 | 0.99±0.00 | 0.99±0.00 | 0.99±0.01 | 0.99±0.00 | 0.99±0.00 | 0.99±0.00 | 0.99±0.00 |

Table 8: Investigation of the oversmoothing effect. Table entries measure the accuracy of the subset of nodes that have a distance to the starting node that the model saw in the training set. Oversmoothing would mean that the introduction of nodes that are further away (and require additional rounds) causes a drop for the close-by nodes. A model that performs achieves the same result across all columns shows resilience against oversmoothing. All models except AMP-RNN demonstrate resilience, though only AMP-ACT and AMP-ITER remain resilient over all graph sizes.

### C.3 OVERSQUASHING

To estimate oversquashing, we compare the performance of the model when it has to solve the shortest path task to when it has to solve three shortest path tasks at the same time on the same graph. This triples the amount of information nodes need to exchange. Figure 8 shows the accuracy for the single problem, Figure 8b the accuracy for the threefold problem. We see that all methods deteriorate in performance, which is expected since the learning task is harder. However, AMP-Iter and AMP-ACT deteriorate least. Since nodes in AMP receive information for one problem at a time, they are less exposed to oversquashing. Recurrent AMP variants perform worse than AMP-Iter and AMP-ACT and even worse than IterGNN and Universal. This suggests that learning to terminate is helpful here.

This task is more difficult than the single source variant because of two reasons: (i) a message can contain $0, 1, 2,$ or $3$ bits of important information (ii) a node cannot terminate after the first important message but needs to wait for three such messages. The second reason aggravates the challenge discussed in the underreaching problem. Additionally, nodes can no longer just set their state based on the message they receive, they also have to take their previous state in account which might contain information about another source. The first reason suggests that messages can carry more information, so blurring that information with that messages of unimportant neighbors makes it harder to find an learn this information. Furthermore, it might be possible that two sources are equal distance, in these cases a synchronous GNN learns of both at the same time and has to account for both. Nodes in AMP receive and can react to the messages individually.

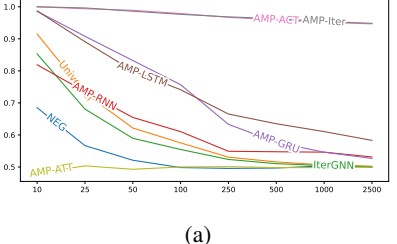

(a)

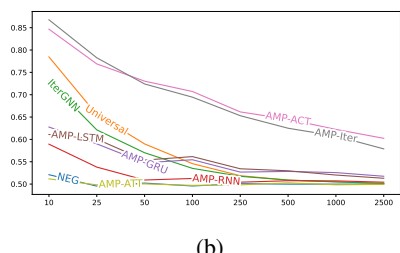

(b)

Figure 8: Accuracy for the shortest path parity task ($y - axis$) for solving (a) one or (b) three tasks at the same time for different graph sizes ($x-$axis). The more accuracy drops between the left to the right figure, the more a method is exposed to oversquashing.

| Model | 10 | 25 | 50 | 100 | 250 | 500 | 1000 | 2500 |
|---|---|---|---|---|---|---|---|---|
| NEG | 0.68±0.16 | 0.56±0.08 | 0.52±0.03 | 0.49±0.02 | 0.49±0.01 | 0.49±0.01 | 0.50±0.01 | 0.49±0.00 |
|  | 0.52±0.03 | 0.49±0.01 | 0.50±0.00 | 0.49±0.00 | 0.50±0.00 | 0.49±0.00 | 0.49±0.00 | 0.50±0.00 |
| Universal | 0.91±0.05 | 0.73±0.05 | 0.62±0.03 | 0.57±0.02 | 0.53±0.01 | 0.51±0.00 | 0.50±0.00 | 0.50±0.00 |
|  | 0.78±0.08 | 0.65±0.06 | 0.58±0.02 | 0.54±0.01 | 0.51±0.00 | 0.50±0.00 | 0.50±0.00 | 0.50±0.00 |
| IterGNN | 0.85±0.11 | 0.68±0.06 | 0.58±0.03 | 0.55±0.02 | 0.52±0.00 | 0.51±0.00 | 0.50±0.00 | 0.50±0.00 |
|  | 0.72±0.07 | 0.62±0.07 | 0.57±0.04 | 0.53±0.02 | 0.51±0.01 | 0.50±0.00 | 0.50±0.00 | 0.50±0.00 |
| AMP-RNN | 0.81±0.16 | 0.73±0.17 | 0.65±0.16 | 0.61±0.14 | 0.54±0.10 | 0.54±0.08 | 0.54±0.07 | 0.53±0.04 |
|  | 0.58±0.04 | 0.53±0.03 | 0.50±0.02 | 0.51±0.03 | 0.50±0.02 | 0.50±0.01 | 0.50±0.00 | 0.50±0.00 |
| AMP-GRU | 0.98±0.01 | 0.90±0.05 | 0.83±0.07 | 0.75±0.10 | 0.63±0.13 | 0.57±0.14 | 0.54±0.10 | 0.52±0.08 |
|  | 0.62±0.03 | 0.58±0.03 | 0.54±0.04 | 0.55±0.04 | 0.52±0.04 | 0.52±0.02 | 0.52±0.02 | 0.51±0.01 |
| AMP-LSTM | 0.98±0.02 | 0.89±0.10 | 0.8±0.10 | 0.74±0.12 | 0.66±0.12 | 0.63±0.11 | 0.61±0.07 | 0.58±0.06 |
|  | 0.61±0.04 | 0.60±0.04 | 0.55±0.04 | 0.56±0.05 | 0.53±0.04 | 0.52±0.02 | 0.52±0.02 | 0.51±0.01 |
| AMP-ATT | 0.48±0.01 | 0.50±0.01 | 0.49±0.01 | 0.50±0.00 | 0.50±0.00 | 0.49±0.00 | 0.49±0.00 | 0.49±0.00 |
|  | 0.51±0.02 | 0.49±0.02 | 0.50±0.01 | 0.49±0.00 | 0.49±0.00 | 0.50±0.00 | 0.49±0.00 | 0.49±0.00 |
| AMP-ACT | 1.0±0.0 | 0.99±0.01 | 0.98±0.01 | 0.97±0.02 | 0.96±0.03 | 0.95±0.04 | 0.95±0.05 | 0.94±0.05 |
|  | 0.84±0.04 | 0.76±0.04 | 0.73±0.03 | 0.70±0.04 | 0.66±0.04 | 0.64±0.04 | 0.62±0.04 | 0.60±0.04 |
| AMP-Iter | 1.0±0.0 | 0.99±0.00 | 0.98±0.03 | 0.97±0.04 | 0.96±0.05 | 0.96±0.06 | 0.95±0.07 | 0.94±0.08 |
|  | 0.86±0.03 | 0.78±0.03 | 0.72±0.02 | 0.69±0.03 | 0.65±0.03 | 0.62±0.02 | 0.60±0.03 | 0.57±0.04 |

Table 9: Investigation for the oversquashing effect. The effect can be seen per method in the degradation from the first to the second row. The first row corresponds to the numbers in Table 2 from the main body. The second row corresponds to the accuracy of solving three shortest path parity tasks at the same time. We only emphasize the best accuracy on this task. AMP-based approaches with self-supervised self termination perform best.

