# OpenReview forum: "Asynchronous Message Passing: A new Framework for Learning in Graphs"
_ICLR.cc/2023/Conference — Submitted to ICLR 2023_

### Official Review · Reviewer_BeiX · 2022-10-20

**Confidence:** 3
**Correctness:** 3
**Technical Novelty And Significance:** 3
**Empirical Novelty And Significance:** 3
**Recommendation:** 6

**Clarity, Quality, Novelty And Reproducibility:**

- Clarity: The paper needs an illustrative example to explain how AMP works on the entire graph. The overall clarity can be improved.

- Quality: The method seems to work with improved expressiveness against 1-WL test.

- Novelty: The idea is novel.

- Reproducibility: The authors provide their experimental code which is well-documented. I believe the results can be reproduced.


**Strength And Weaknesses:**

### Strengths

- The idea of AMP is interesting, especially its various variants of designs.

- The results on the synthetic datasets are impressive.

- The provided experimental code is well-documented. The environment, package dependency, and the steps to reproduce the results are all stated.

### Weaknesses

- The clarity of the paper can be greatly improved. A pictorial illustration of the proposed framework would help a lot. I cannot understand how AMP works on the entire graph/graphs.

- The clarity of the pseudo-code (Algorithm 1) can be improved. Please specify the input and output of the algorithms. Also, what is $n$ in the for loop? Does Algorithm 1 works for one node or the entire graph?

- The running time of AMP is poor as it can only run on a single CPU. I do appreciate the authors pointing this limitation out.

- The AMP performance on real-world graph classification benchmark datasets still falls behind the state-of-the-art methods by a large margin. Also, the reported results for baseline methods seem to contradict the previous works.

### Detail comments

While I really want to learn what AMP is and how it can separate any pair of graphs in the optimal case, the clarity of the paper prevents me to understand AMP thoroughly. For example, I can understand how the asynchronous works for an isolated node from Figure 1. However, I cannot understand how it works for the entire graphs. When the constant delay is used, does the initial message simply the node features for each node? Do these initial messages happens at the same time or do we start the propagation from one node? The description of the AMP framework is too vague for me to understand. I suggest the authors add a pictorial example of the AMP framework on a simple graph. On the other hand, Algorithm 1 should state the inputs, outputs and hyperparameters. I am not sure what $n$ stands for in the for loop, where I suspect it to be the current node. I hope the authors can elaborate more about the AMP framework.

Although the authors show that AMP is very powerful and expressive, it does not have time complexity analysis nor demonstrates a reasonable running time. Note that the brute force method can also resolve the graph isomorphism test, it just takes beyond polynomial times. Hence, merely demonstrating the expressiveness of the method is not satisfactory, especially with the claim of being able to solve a notoriously hard problem (i.e. graph isomorphism test). I would suggest the authors report the running time compared with baseline GNNs. I also wish the authors can compare the time complexity of AMP against the standard message passing framework.

Regarding the experiment section, I find it awkward that there is no single model under AMP framework that can simultaneously do well on all three experiments. In section 5.1 the authors compare AMP-C and AMP-R with baseline methods, where the tasks are synthetic node and graph classifications. However, the authors compare different variants AMP-RNN, AMP-ACT, AMP-Iter…etc with baseline methods in section 5.2 when predicting whether the shortest path of the graph is even or odd. Then on real-world benchmark datasets, the authors only show the result of AMP-RNN. Why do we need so many variants? How do we know that the superior performance of the result indeed mainly comes from AMP instead of these sophisticated and specialized variant designs? Finally, according to the ESAN paper by Bevilacqua et al. 2022, the state-of-the-art results on five tested benchmark datasets are close to ESAN. However, the authors show that GCN can significantly outperform ESAN. I wonder why such an inconsistency exists.


**Summary Of The Paper:**

The authors propose an asynchronous message passing (AMP) framework which is theoretically more powerful than the standard message passing framework. The experiments demonstrate its superior performance on synthetic datasets and moderate performance on graph classification benchmark datasets with small sizes.

**Summary Of The Review:**

The authors propose the asynchronous message passing (AMP) framework which seems to resolve many existing issues of standard message passing based methods. The experiment results on synthetic data demonstrate the superior performance of AMP and moderate performance on real-world benchmark datasets. My major concerns of the paper are its clarity and time complexity. Some other concerns pertaining to the experiments are also listed. I hope the authors can elaborate more about the AMP framework and address my concerns listed above. I am happy to raise my score if my concerns are well addressed.

==================post rebuttal========================

I thank the authors' rebuttal. My major concerns are addressed and hence I increase my rating from 5 to 6.

---

> ### Author Response · Authors · 2022-11-15
> **Response to reviewer BeiX 2/2**
>
> > Regarding the experiment section, I find it awkward that there is no single model under AMP framework that can simultaneously do well on all three experiments. In section 5.1 the authors compare AMP-C and AMP-R with baseline methods, where the tasks are synthetic node and graph classifications. However, the authors compare different variants AMP-RNN, AMP-ACT, AMP-Iter…etc with baseline methods in section 5.2 when predicting whether the shortest path of the graph is even or odd. Then on real-world benchmark datasets, the authors only show the result of AMP-RNN. Why do we need so many variants? How do we know that the superior performance of the result indeed mainly comes from AMP instead of these sophisticated and specialized variant designs? Finally, according to the ESAN paper by Bevilacqua et al. 2022, the state-of-the-art results on five tested benchmark datasets are close to ESAN.
>
> This is bad naming on our part, actually the AMP-C model for the first set of experiments equals the AMP-RNN model. We will adapt the naming calling these models AMP-RNN-C and AMP-RNN-R.
>
> We decided to choose AMP-RNN for all benchmarks since it is a simple instance in the AMP framework (no $\rho$ network, simple feedforward layers for the remaining functions). We would argue that we can attribute the performance of AMP-RNN largely to the AMP framework. AMP-RNN performs very well in Table1 and comparable to GNNs in Table 2, even slightly outperforming them on much larger graphs. In Table3 we can see that this simple AMP version performs comparable to simpler GNNs such as GAT, GIN or GraphSAGE. However, performance is not on par with stronger GNNs such as ESAN, PPGN, or DropGNN (and surprisingly, GCN).
>
> On the other hand, we also wanted to highlight that AMP is not a single network but a framework in which we can realize many possible networks (similar to how synchronous message passing GNNs as a framework encompass GCN, GIN, and many more). This is why we presented this zoo of alternative AMP models. Since AMP works close to perfect on the expressiveness benchmark and the runtime issues make experimenting on the graph classification datasets difficult,  we presented alternatives for the shortest path problem.

---

> > ### Comment · Reviewer_BeiX · 2022-11-22
> > **Re:**
> >
> > I thank the authors for their response. My main concerns are addressed, hence I increase my rating accordingly.
> >
> > On the other hand, while the authors demonstrate the time complexity of the forward pass of AMP and standard GNNs are the same, I wonder if the sequential nature of the AMP makes its backward pass harder to compute (i.e., higher computational time)? Since computing the backward pass in AMP now also depends on the delay, hence in the worst case, it is possible that nothing can be parallel For example, one node at most received at most $k$ messages, and in the worst case, they will cascade. This results in the backward propagation need to process $O(kn)$ messages for one node ***sequentially***. This might also be the bottleneck that prevents AMP to be efficient. Maybe I misunderstand something though.

---

> ### Author Response · Authors · 2022-11-15
> **Response to reviewer BeiX 1/2**
>
> > The clarity of the pseudo-code (Algorithm 1) can be improved. Please specify the input and output of the algorithms. Also, what is in the for loop? Does Algorithm 1 works for one node or the entire graph?
>
> Thanks for letting us know, we tried to strip away all but the most core parts but we can see now that we over-shortened. The forward pass AMP is as synchronous GNN variants. It takes a graph and the node feature matrix X. At the end of this algorithm we have new states for every node (derived from X and the graph structure) that we could use for node level classification. Or we could pool them and perform graph classification. We’ll expand the pseudocode to show both variants. This will also clarify that this algorithm works on the entire graph and not just a single node.
>
> We also added a visual example of Algorithm one to section 3. Is this what you were looking for and did it help clear up confusions about AMP?
>
> > The running time of AMP is poor as it can only run on a single CPU. I do appreciate the authors pointing this limitation out.
>
> Yes, we added an analysis of the complexity and found that AMP and GNNs are similar. It seems the problems come from this limitation. We are hoping to overcome it with multiprocessing (multithreading does not really work in Python). There seems to exist a pytorch module to also support backpropagation.
>
> > Also, the reported results for baseline methods seem to contradict the previous works.
> > However, the authors show that GCN can significantly outperform ESAN. I wonder why such an inconsistency exists.
>
> We were equally surprised by GCN’s results. Searching in the literature, we did not find results on those datasets for GCN after the initial GIN paper (Xu et al. How powerful are graph neural networks). There, the authors mention that they did not run exactly GCN but an architecturally-close 1-layer variant. We did not find later papers that compared using these graph classification datasets  and compared against GCN.
>
> In our implementation, we copied the original code for GIN and swapped in pytorch geometric’s implementation for GCN and ran with the usual hyperparameter options. We got those options without any further effort or tuning.
>
> > While I really want to learn what AMP is and how it can separate any pair of graphs in the optimal case, the clarity of the paper prevents me to understand AMP thoroughly.
>
> Thanks for this feedback. We will expand the algorithm as described above. We also like the suggestion to show AMP’s execution over time. To this end, we will discuss a variant of Figure1 unrolled through time in section 3.
>
> > Although the authors show that AMP is very powerful and expressive, it does not have time complexity analysis nor demonstrates a reasonable running time. Note that the brute force method can also resolve the graph isomorphism test, it just takes beyond polynomial times. Hence, merely demonstrating the expressiveness of the method is not satisfactory, especially with the claim of being able to solve a notoriously hard problem (i.e. graph isomorphism test). I would suggest the authors report the running time compared with baseline GNNs. I also wish the authors can compare the time complexity of AMP against the standard message passing framework.
>
> We added an analysis of the complexity in the general response, it shows that AMP has comparable complexity to GNNs. We also incorporated reviewer 6FHJ to more-precisely investigate the graph isomorphism. AMP requires $O(\frac{n}{2})$ space so there is no polynomial solution. Nevertheless, AMP solving graph isomorphism is a strong positive result. When analyzing expressive GNNs we often find some $k$ such that the GNN cannot separate graphs that the $k$-WL test also fails on. AMP solving general graph isomorphism means that such a $k$ does not exist for AMP.

---

### Official Review · Reviewer_oodJ · 2022-10-24

**Confidence:** 3
**Correctness:** 3
**Technical Novelty And Significance:** 2
**Empirical Novelty And Significance:** 3
**Recommendation:** 5

**Clarity, Quality, Novelty And Reproducibility:**

I find that the paper lacks clarity in that it discusses various arguments for asynchronous message passing at length (human communication, recurrent architectures, GNNs, etc.) without making a point and giving a clear and concise motivation. Therefore it is also hard to gauge the actual contribution. The fact that the proofs are only presented in the appendix (without any summary of the main proof idea) does not help either. Even going through the proof of Theorem 3.1 it does not become obvious which parts are actual contributions of the paper and which ones of Awerbuch (1985).
It also seems that the references are not comprehensive. Asynchronous message passing has for example been successfully applied in traditional message passing algorithms (see e.g., [1,2,3]). The work in [4] shows the benefits of asynchronous event-based GNNs and is closely related.

Minor edit: "We hypothesize lies in better algorithmic alignment" is not a complete sentence

[1] Elidan, Gal, et al. "Residual belief propagation: Informed scheduling for asynchronous message passing." arXiv preprint arXiv:1206.6837 (2012).

[2] Knoll, Christian, et al. "Message scheduling methods for belief propagation." Joint European Conference on Machine Learning and Knowledge Discovery in Databases. Springer, Cham, 2015.

[3] Aksenov, Vitalii, Dan Alistarh, and Janne H. Korhonen. "Scalable belief propagation via relaxed scheduling." Advances in Neural Information Processing Systems 33 (2020): 22361-22372.

[4] Schaefer, Simon, Daniel Gehrig, and Davide Scaramuzza. "AEGNN: Asynchronous Event-based Graph Neural Networks." Proceedings of the IEEE/CVF Conference on Computer Vision and Pattern Recognition. 2022.

**Strength And Weaknesses:**

The strength of the paper is that it proposes a potential remedy against certain problems of GNNs as over smoothing and underreaching. This is done by a relatively straightforward and intuitive algorithm

Yet, there are certain limitations to the work in its current form:
* The literature is not conclusive and a more focused presentation would improve the paper overall (see clarity,... below)
* As the authors say themselves, the proposed approach is not quite state-of-the-art. While this is not a problem as such (if the underlying idea is promising) it does not scale well because of the runtime. Overall, I have the feeling that AMP is a neat idea that, however, is not yet ready and would still require significant work.
* The theoretical contribution is overstated; i.e., it is not clear that AMP is at least as powerful as a GNN. Just because it can represent the same things in principle, it is not clear if and how this representation can be obtained.
* I really miss an honest and thorough discussion of the experiments and see this as a largely missed opportunity to understand AMP better. Many statements seem to come out of the air and are not corroborated. E.g.,
** It is claimed that AMP/R produces mediocre results because of unstable gradients. Why are the gradients unstable, what could be done against it, what is special for AMP in that respect?
** It is claimed that further improvements in AMP will reach a competitive performance. How can you be so sure? Which aspects limit the performance right now?. How could these aspects be improved?
** There are nice plots regarding the effect of underreaching and oversmoothing in the Appendix. Unfortunately, these sections do not actually analyze the experiments, but only state the results.




**Summary Of The Paper:**

This paper aims to improve shortcomings of GNNS by updating the messages in an asynchronous manner. The paper shows how this relates to the standard GNN and argue how it can be used for testing graph isomorphism.

**Summary Of The Review:**

The paper makes an interesting contribution from a conceptional point of view. It does not make a good argument for the proposed approach though, since a) the literature is not conclusive, b) it does not scale to larger graphs, c) the amount of theoretical contribution is not clear, and d) a thorough discussion and interpretation of the experiments (and thus the potential benefits and problems) is missing.

---

> ### Author Response · Authors · 2022-11-15
> **Response to revieer oodJ 2/2**
>
> > I find that the paper lacks clarity in that it discusses various arguments for asynchronous message passing at length (human communication, recurrent architectures, GNNs, etc.) without making a point and giving a clear and concise motivation.
>
> We will try to reformulate the arguments in the introduction and method introduction to make this more clear. In the core, the contribution of AMP is that it is a neural network framework for graphs that does not send all messages at once but handles messages individually. To the best of our knowledge this is the first framework doing so (for example, AEGNN does restrict processing to certain parts of the graph but updates there follow the synchronous approach) Another perk of AMP is to start computation locally but that idea exists, for example, also in AEGNN.
>
> The fact that the proofs are only presented in the appendix (without any summary of the main proof idea) does not help either. Even going through the proof of Theorem 3.1 it does not become obvious which parts are actual contributions of the paper and which ones of Awerbuch (1985).
>
> Thanks for the feedback that we cut that section too short. We will try to make space to give main proof ideas in the main body.
>
> With regards to the proof for 3.1: the proof framework (having safe nodes and progressing when nodes are safe) is that of Awerbuch. However, the original idea is for the LOCAL model where we can have arbitrary local computation. The contribution of our proof is showing that we can derive a similar protocol in the framework of Algorithm 1, that works exactly (and not, for example, approximated to some degree via an argument with the universal approximation theorem) and with reasonable overhead (constant more space in the nodes, and linear more messages (additional acks and noops for every neighbor state received).
>
> > Asynchronous message passing has for example been successfully applied in traditional message passing algorithms (see e.g., [1,2,3]). The work in [4] shows the benefits of asynchronous event-based GNNs and is closely related.
>
> Thanks for the literature pointers. The works in the field of probabilistic graphical models are very interesting: individual handling of messages also improved over synchronous computation.. The recent work on AEGNN also fits the paper nicely. We will add all papers to the related work.

---

> > ### Comment · Reviewer_oodJ · 2022-11-22
> > **Acknowledge response**
> >
> > I've read the response and thank the authors for clarifying certain points and improving certain aspects of the paper. I've changed my score accordingly.
> >
> > While I agree that the question of what AMP can represent in principle is of course an important one, I still think that certain limitation of what can actually be learned are important to discuss. This becomes particularly obvious as we consider for example the prohibitive runtime for large graphs. The paper definitely has potential, and I am convinced that it will make for a strong submission if the proposed method will scale a bit better and if its practical aspects are thoroughly discussed.

---

> ### Author Response · Authors · 2022-11-15
> **Response to reviewer oodJ**
>
> Thank you for your review
>
> > The theoretical contribution is overstated; i.e., it is not clear that AMP is at least as powerful as a GNN. Just because it can represent the same things in principle, it is not clear if and how this representation can be obtained.
>
> We agree that results which representations AMP can practically learn would be superior than finding out which AMP can represent (but maybe not learn). Nevertheless, the latter results are still important to understand the limits of individual methods. We think that it should be helpful to understand what AMP potentially understand, when we want to investigate which of these representations it can learn.
>
> > It is claimed that AMP/R produces mediocre results because of unstable gradients. Why are the gradients unstable, what could be done against it, what is special for AMP in that respect?
>
> In every forward pass of AMP-R, messages delays are randomly sampled. This means that the order in which messages arrive changes, which can also change the states of nodes (for example, both $v_1$ and $v_2$ send a message to $v_3$ but in one run $v_1$’s message arrives first, in another run $v_2$’s message. This can lead to different states for $v_3$.). After each run we backpropagation computes an update to the neural network weights given the node states of this run. However, it is not clear that updates for one set of states also benefits the states of the next run.
>
> One potential remedy is training longer, since over many steps, we will see messages frequencies according to their probabilities and the gradients can optimize for the expected message sequence. We assume that averaging over multiple runs should also help, again to create a representative average message sequence.
>
> > It is claimed that further improvements in AMP will reach a competitive performance. How can you be so sure? Which aspects limit the performance right now?. How could these aspects be improved?
>
> Our main limitation is the slow runtimes on which we are working. This prevents exploring the space of alternative more sophisticated AMP models on these datasets. So far, we experimented with AMP-RNN which is one of the more simple models in the AMP framework. Because of the results in Table 2 and the improvements over AMP-RNN, we have hope that there are models in the AMP framework that will also improve upon AMP-RNN for these datasets.
>
> > There are nice plots regarding the effect of underreaching and oversmoothing in the Appendix. Unfortunately, these sections do not actually analyze the experiments, but only state the results.
>
> Thanks for the suggestions which aligns with the remark from reviewer 6FHJ. We expanded the appendix section to give some more intuition why AMP might be less exposed to effects such as underreaching and oversmoothing, which we want to quickly sketch here:
> Underreaching: For a node with distance $d$ and degree $k$ to receive a message that allows correct classification it needs to receive $d\cdot k$ messages in a GNN on the graph size) but only $k$ messages in AMP.
> Oversmoothing: A node in a GNN needs to execute at least $d$ many updates which aggregate over its neighborhood and can contribute to oversmoothing. Nodes in AMP listen to fewer messages.
> Oversquashing: In the iteration a node receives the message that allows classification aggregated with $k-1$ other unimportant messages, while the same node in AMP receives the important message in isolation. This effect is stronger with higher node degrees.

---

### Official Review · Reviewer_qHED · 2022-10-25

**Confidence:** 3
**Correctness:** 4
**Technical Novelty And Significance:** 3
**Empirical Novelty And Significance:** 2
**Recommendation:** 6

**Clarity, Quality, Novelty And Reproducibility:**

I found it a little bit difficult to follow the paper. It would be more easier to be followed if more intuitive description and direct mathematical equations can be provided.

**Strength And Weaknesses:**


Strengths:

1. The motivation and analysis of proposing AMP is inspiring and novel.

2. Treating messages individually is a novel and reasonable idea to go beyond the widely used message aggregation schema. This idea is novel and could be inspiring to the community.

3. The expressive analysis of AMP can answer the question why we should use AMP compared to message passing GNNs.

4. The empirical performance on the expressiveness synthetic benchmarks are compelling and can support the main claim well.

Weakness:

1. The empirical comparison on more and lager real benchmarks should be provided. In the current experimental results, only Table 3 is for real graphs. However, the size and number of the graphs are limited and cannot support the method strongly. I highly recommend performing more experiments on larger graph datasets.

2. The proposed new framework seems to be inefficient. The theoretical and/or empirical analysis of the complexity should be included.


**Summary Of The Paper:**

This paper proposes a new framework, asynchronous message passing (AMP), for graph learning. To be specific, each message passed along the edges is treated individually, which differs from the widely used message passing GNNs. Also, the expressiveness of AMP has been characterized. Experiments on synthetic datasets and real benchmarks are performed to evaluate AMP.

**Summary Of The Review:**

Overall, I think this work has merits in terms of the motivation and method. On the other hand, more analysis and experiments should be added.

---

> ### Author Response · Authors · 2022-11-15
> **Response to reviewer qHED**
>
> Thank you for your review.
>
> > The empirical comparison on more and lager real benchmarks should be provided. In the current experimental results, only Table 3 is for real graphs. However, the size and number of the graphs are limited and cannot support the method strongly. I highly recommend performing more experiments on larger graph datasets.
>
> We agree with this that experimenting on larger graphs would contribute to the paper. Currently, the technical limitations around limited parallel processing prevents testing on larger datasets but we hope to alleviate this with a multiprocessing solution.
>
> > The proposed new framework seems to be inefficient. The theoretical and/or empirical analysis of the complexity should be included.
>
> Thanks for the suggestion. Since multiple reviews brought this up we bundled a complexity analysis in a general response at the top and added it to the revised paper version in section 3.3.
>
> In terms of complexity, AMP and GNNS send comparably many messages and do comparably many node updates. The differences in runtime come from technical reasons: AMP unlike GNNs is not as inherently parallel and cannot exploit the acceleration brought by GPUs. We also discuss this in more detail in the general response.
>
> > I found it a little bit difficult to follow the paper. It would be more easier to be followed if more intuitive description and direct mathematical equations can be provided.
>
> Are there (sub-)sections that sprang to your mind? We are working on a revised version to incorporate suggestions e.g., complexity and happy to make more changes to improve clarity. We already added a new example run Algorithm 1 in section 3, and added more explanation to the results in Appendix C.

---

> > ### Comment · Reviewer_qHED · 2022-11-27
> > **Thanks**
> >
> > Thanks for the response.
> >
> > I highly encourage to strengthen the experimental part. If we cannot increase the size of graphs due to scalability issue, more results on datasets with more small graphs should be added. The numbers of graphs in the used datasets in Tables are too small, thus being less convincing

---

### Official Review · Reviewer_6FHJ · 2022-10-25

**Confidence:** 4
**Correctness:** 2
**Technical Novelty And Significance:** 3
**Empirical Novelty And Significance:** 2
**Recommendation:** 5

**Clarity, Quality, Novelty And Reproducibility:**

**Clarity:** This paper is hard to follow.
I have some questions as follows.
1. How does the reaction block distinguish the neighbors with the same node feature and rooted subtree? Assume that node $v_1$ receives the messages from its neighbors $v_2$ and $v_3$ with the same node feature and rooted subtree. For an asynchronous GNN, node $v_1$ first receives message $m_1$ from $v_2$ and then sends the same message $m_2$ to $v_2, v_3$ according to Steps 8,9 in Algorithm 1. The updated node embeddings of $v_2$ and $v_3$ (denoted by $z_2$ and $z_3$ respectively) are still the same due to the same received message. Next, nodes $v_2, v_3$ generate the messages $m_3$ and $m_4$ with $m_3=\mu(z_2, m_2)= \mu(z_3, m_2)=m_4$, where $\mu$ is the message function (Step 9 in Algorithm 1). If node $v_1$ sequentially receives messages $m_3, m_4$ from $v_2, v_3$, how does the reaction block discard $m_3$ and keep $m_4$ in the proof for Theorem 3.1, such that AMP simulates synchronous GNNs, for which node $v_1$ receives and keeps a single message from $v_2, v_3$ respectively?
2. What is the meaning of "taking" in Table 6? Is it a state or message?
3. What is the expression of the delay distribution $\mathcal{D}$ in the experiments?
4. In the third paragraph of the proof for Theorem 3.1 (A.1), the number of messages that the node is waiting on may be "w".
5. In the proof for Lemma 4.2 (A.2), "FOUND-(j+k)" may be "FOUND-(j+i)".

**Quality:** The authors may want to provide a rigorous theoretical analysis.

**Novelty:** This proposed AMP framework is interesting.

**Reproducibility:** The authors provide the codes for reproducibility.


**Strength And Weaknesses:**

**Strengths:**
1. This proposed AMP framework is interesting.
2. Experiments demonstrate that the AMP-based model outperforms powerful synchronous GNNs on several GNN expressiveness benchmarks in terms of accuracy.

**Weaknesses:**
1. The authors may want to provide a rigorous proof for Theorem 4.7. Some weaknesses are as follows.
    1. In the proof, the authors make the assumption of GNN's width and depth, which is not mentioned in Theorem 4.7.
    2. Corollary 3.1 in [1] assumes that the width of GNNs is unbounded, which is not equivalent to the assumption of a sufficiently wide GNN.
    3. The authors may want to provide the lower bound of the depth of GNNs.
    4. The application of Corollary 3.1 is confusing. What is the definition of Turing computable function in Corollary 3.1 in [1]? If a GNN can compute any Turing computable function, how to show that it can separate any pair of graphs?
2. From Table 3, we can see that the accuracy of AMP is clearly lower than existing powerful GNNs (e.g., ESAN). The authors claim that the low accuracy is due to little investigation into AMP architectures and hyperparameters, but it is not the key reason in my opinion, as more AMP architectures with more hyperparameters are available to ensure a fair comparison for the authors.
3. The authors may want to compare the time complexity and runtimes of AMP with those of synchronous GNNs.
4. The authors claim that AMP can propagate messages over large distances in graphs without the corresponding theoretical analysis.

[1] Andreas Loukas. What graph neural networks cannot learn: depth vs width. In International Conference on Learning Representations (ICLR), 2020.


**Summary Of The Paper:**

The authors propose the asynchronous message passing (AMP) framework to enhance the expressiveness of graph neural networks. In the AMP framework, nodes receive messages from their neighbors individually. The authors try to show that AMP is more powerful than the 1-WL test and can even compute graph isomorphism. Experiments demonstrate that the AMP-based model outperforms powerful synchronous GNNs on several GNN expressiveness benchmarks in terms of accuracy.

**Summary Of The Review:**

I recommend weak reject due to the concern about the theoretical analysis, the experiments, and the writing (See Weaknesses and Clarity). If the authors can properly address my concerns, I am willing to raise my score.

---

> ### Author Response · Authors · 2022-11-15
> **Response to reviewer 6FHJ 2/2**
>
> > What is the meaning of "taking" in Table 6? Is it a state or message? In the third paragraph of the proof for Theorem 3.1 (A.1), the number of messages that the node is waiting on may be "w". In the proof for Lemma 4.2 (A.2), "FOUND-(j+k)" may be "FOUND-(j+i)".
>
> Thanks, these are all correct and adapted. Taking was a synonym for ‘claiming’ from an older version.
>
> > What is the expression of the delay distribution in the experiments?
>
> AMP-R samples from a normal distribution with mean and standard distribution 0.5. The lower bound for delays is 1e-10.
>
> > How does the reaction block distinguish the neighbors with the same node feature and rooted subtree?
> Given your example, it is not possible for $v_1$ to discard $m_3$ and keep $m_4$, assuming that $m_3$ arrives before $m_4$. However, it is possible to keep the earlier message $m_3$ and discard the later message $m_4$. The discard function $\rho$ takes the current state of $v_1$ as a parameter. Let us assume $v_1$ in state $s$ receives message $m_3$ and $\rho(s, m_3=m_4)$ decides to keep the message. As part of the update, $v_1$ transitions to a state $s’$. When $m_4$ arrives next, the discard function $\rho(s’, m_3=m_4)$ using the new state $s’$ might choose to discard the second-arriving message.

---

> > ### Comment · Reviewer_6FHJ · 2022-11-17
> > **Thanks for the authors' response.**
> >
> > Thanks for the authors' response. However, my major concerns 1-4 and Question 1 in the Clarity part have not been properly addressed. I have some questions about  Concerns 1, 3, 4 and Question 1 as follows.
> > 1. (Concern 1) Is $G = (V_1\cup V_2, E_1\cup E_2)$ where $G_1 = (V_1, E_1)$ and $G_2 = (V_2, E_2)$? If so, then $G$ is a disconnected graph, whose diameter $\delta_G$ is unbounded.
> > 2. (Concern 3) Why do the complexities of EASN and DropGNN in the revised version differ from [1] and [2]? I suggest the authors analyzing the complexity following EASN[1] and DropGNN[2].
> > 3. (Question (1) in Clarity) I suggest the authors providing an example about how AMP simulates a given GNN (Theorem 3.1) to avoid misunderstanding. Specifically, I still have the following question.
> > 	1. The proposed protocol in the rebuttal is difficult to apply to another example. In the rebuttal, the authors claim that the proposed protocol is to keep $m_3$ and discard $m_4$ for node $v_1$. Let us consider another example where the feature of $v_2$ is different from $v_3$, i.e., $z_2 \neq z_3$. According to Step 9 in Algorithm 1, we have $m_3 \neq m_4$. Node $v_1$ can not know that $z_2 \neq z_3$ and $m_3 \neq m_4$ before receiving $m_4$, as $v_1$ does not receive any messages from $v_3$. If we apply the protocol in the rebuttal to the example, then the protocol is still to keep $m_3$ and discard $m_4$ for node $v_1$, such that node $v_1$ receives and keeps twice messages from $v_2$ rather than a single message from $v_2,v_3$ respectively. If my derivation is correct, then AMP can not simulate synchronous GNNs---for which node $v_1$ receives and keeps a single message from $v_2,v_3$ respectively---in the example.
> > 4. (Concern 4) The authors may want to include the analysis of AMP's ability to propagate long-term messages in the main text, as they claim that it is one of the main contributions.
> >
> >
> > [1] Equivariant Subgraph Aggregation Networks. In International Conference on Learning Representations (ICLR), 2022.
> >
> > [2] DropGNN: Random Dropouts Increase the Expressiveness of Graph Neural Networks. In Conference on Neural Information Processing Systems (NeurIPS), 2021.

---

> > > ### Author Response · Authors · 2022-11-18
> > > **An example simulation of the protocol for proof 3.1**
> > >
> > > > I suggest the authors providing an example about how AMP simulates a given GNN (Theorem 3.1) to avoid misunderstanding.
> > >
> > > That is a good idea, let us run an example on Let us run the Theorem 3.1 on the example graph G = ({$v_1$, $v_2$, $v_3$, $v_4$}, {{$v_1, v_2$}, {$v_1, v_3$}, {$v_2, v_4$}, {$v_3, v_4$}}). We track the states of the four node in a vector $((s_1, a_1, w_1, u_1, l_1),(s_2, a_2, w_2, u_2, l_2),(s_3, a_3, w_3, u_3, l_3), (s_4, a_4, w_4, u_4, l_4))$.
> > >
> > > Initially, the states are $((s_1, 0, 1, 2, 1, L),(s_2, 0, 1, 2, 1, L),(s_3, 0, 1, 2, 1, L),(s_4, 0, 1, 2, 1, L))$. Let us send the initial message to $v_2$. This causes the transition to
> > > $((s_1, 0, 1, 2, 1, L),(s_2, 0, 1, 2, 0, L),(s_3, 0, 1, 2, 1, L),(s_4, 0, 1, 2, 1, L))$. Then $v_2$ and sends a PULSE message to $v_1$ and $v_4$.
> > >
> > > Let us assume $v_1$ receives the message first.Then $v_1$ transitions to node states $((s_1, s_2, 0, 2, 0, L),(s_2, 0, 1, 2, 0, L),(s_3, 0, 1, 0, 1, L), (s_4, 0, 1, 0, 1, L))$ and sends a pulse to both $v_2$ and $v_3$.
> > >
> > > Next, $v_2$ receives the message from $v_1$. Since $v_2$ transitioned to another state already, $i$ is not $1$ anymore, and $v_2$ updates with the sixth equation from Table 4 leading to $((s_1, s_2, 0, 2, 1, L),(s_2, s_1, 0, 2, 0, L),(s_3, 0, 1, 0, 0  L), (s_4, 0, 1, 0, 0, L))$. As part of updating with the sixth equation $v_2$ sends a NOOP, which will be ignored by its neighbors. On the other hand, $v_3$ is still in a state where $i=1$. Upon receiving the message by $v_1$, it updates with the fourth equation to $((s_1, s_2, 0, 2, 1, L),(s_2, s_1, 0, 2, 1, L),(s_3, s_1, 0, 2, 0, L), (s_4, 0, 0, 2, 1, L))$ and sends a PULSE to $v_1$ and $v_4, which will not be discarded.
> > >
> > > At this point, we have recreated the above scenario where $v_2$ sends twice to $v_1$; node $v_3$ sent one message. Node $v_1$ keeps one of these messages each. Let us now continue a bit longer.
> > >
> > > Let $v_1$ now receive the PULSE message from $v_3$. This is the last neighbor missing for $v_1$ to compute the actual update with the fifth equation.
> > > $((s_1’, 0, 1, 1, 0, L-1),(s_2, s_1, 0, 2, 0, L),(s_3, s_1, 0, 2, 0, L), (s_4, 0, 1, 2, 1, L))$ with $s_1’=UPDATE(s,a+m$). Therefore, $s_1'$ matches the state of $v_1$ after the first GNN layer. Now, $v_1$ can send ACK to its neighbors that it received and processed their pulses. Nodes $v_2$ and $v_3$ receive those and update to $((s_1’, 0, 1, 1, 0, L-1),(s_2, s_1, 0, 1, 0, L),(s_3, s_1, 0, 1, 0, L), (s_4, 0, 1, 2, 1, L))$.
> > >
> > > The two outstanding messages are the messages from $v_2$ and $v_3$ to $v_4$. Let the one from $v_2$ arrive first. Node $v_4$ will transition to
> > > $((s_1’, 0, 1, 1, 0, L-1),(s_2, s_1, 0, 1, 0, L),(s_3, s_1, 0, 1, 0, L), (s_4, s_2, 0, 2, 0, L))$ and send a pulse to both $v_2$ and $v_3$.
> > >
> > > Let $v_3$ receive this pulse next after which it can also update its state to $((s_1’, 0, 1, 1, 0, L-1),(s_2, s_1, 0, 1, 0, L),(s_3’, 0, 1, 0, 0, L), (s_4, s_2, 0, 2, 0, L))$ and send an ACK. Node $v_1$ receives the ACK and transitions to $((s_1’, 0, 1, 0, 0, L-1),(s_2, s_1, 0, 1, 0, L),(s_3’, 0, 1, 1, 0, L), (s_4, s_2, 0, 2, 0, L))$.
> > >
> > > Now $v_1$ (as $v_3$) is ready to emit a new PULSE on the next non-NOOP message (which will be an ACK from $v_2$). Next, $v_2$ could receive the pulse from $v_4$ and update with the fifth equation to  $((s_1’, 0, 1, 0, 0, L-1),(s_2', 0, 1, 0, 0, L),(s_3’, 0, 1, 0, 0, L), (s_4, s_2, 0, 2, 0, L))$ and send an ACK. Upon receiving this ACK in $v_1$, $v_1$ updates to $((s_1’, 0, 1, 2, 0, L-1),(s_2', 0, 1, 0, 0, L),(s_3’, 0, 1, 1, 0, L), (s_4, s_2, 0, 2, 0, L))$ and triggers a new PULSE.
> > >
> > > Node $v_3$ (and/or $v_2$) could receive this pulse next and update to $((s_1’, 0, 1, 2, 0, L-1),(s_2', s_1’, 0, 0, 0, L),(s_3’, s_1', 0, 0, 0, L), (s_4, s_2, 0, 2, 0, L))$ and send a NOOP. Before node $v_4$ does not finish simulating the previous layer and send an ACK, neither $v_2$ nor $v_3$ will send a new PULSE allowing $v_1$ to progress again.

---

> > > > ### Comment · Reviewer_6FHJ · 2022-11-28
> > > > **Thanks for the authors' response.**
> > > >
> > > > Thanks for the authors' response. The response has addressed my Concerns 1, 4.
> > > >
> > > >
> > > > However, the response does not properly address Question 1 in Clarify and Concerns 2, 3 in Weaknesses. The reasons are as follows.
> > > > 1. (Question 1 in Clarity) The provided example in the response is confusing, as it contradicts the proof  Theorem 3.1.
> > > >     - The authors claim that the states $((s_3,0,1,2,1,L), (s_4,0,1,2,1,L))$ of nodes $v_3,v_4$ become $((s_3,0,1,0,1,L), (s_4,0,1,0,1,L))$ after $v_1$ receives the PULSE message from $v_2$ in the response. However, the 4-th component of the state can not become "0" from "2" according to Table 4 (Column $u'$).
> > > >     - The authors claim that the states $((s_1,s_2,0,2,0,L), (s_2,0,1,2,0,L))$ of nodes $v_1,v_2$ become $((s_1,s_2,0,2,1,L), (s_2,s_1,0,2,1,L))$ after $v_2$ receives the message from $v_1$ in the response. However, the 5-th component of the state can not become "1" from "0" according to Table 4 (Column $i'$).
> > > >     - The authors may want to point out that the type of the initial message is "origin" in the proof for Theorem 3.1.
> > > >     - The initial state in the proof may be $(x,0,D-1,D,1,L)$ according to the provided example.
> > > > 2. (Concern 2 in Weaknesses) In my previous comments, I have pointed out that the little investigation into AMP architectures and hyperparameters is not the key reason for the low accuracy of AMP. The authors agree with me, but they say they can not deal with it due to the limited time.
> > > > 3. (Concern 3 in Weaknesses) The time complexity of AMP ignores some constants (e.g., $k$ in the response), which indicates the additional costs of AMP. The authors may want to compare the runtime of AMP with that of synchronous GNNs---which I have suggested in my previous comments---to demonstrate that the constants are unimportant.
> > > > 4. (Concern 3 in Weaknesses) The time complexity of the update step in AMP may be $O(n \Delta_{max})$, as each node updates its state by a single message from its neighbors (Step 8 in Algorithm 1) rather than all messages from its neighbors.
> > > >
> > > >
> > > > Overall, I think this paper is interesting, while it is not ready to publish given its current status. The authors may want to carefully rewrite their technical derivations (see Question 1 in Clarity). I suggest the authors further improving the experiments (see Concerns 2, 3 in Weaknesses) and the writing of this paper.

---

> > > ### Author Response · Authors · 2022-11-18
> > > **Response to the concerns**
> > >
> > > Thanks for the reply, there were indeed some missing information and errors.
> > >
> > > Concern1:
> > > Good catch, thank you. We need to connect the two graphs to have a bounded diameter and Theorem 3.1 of Loukas also requires a connected graph. We can address this by using the graph $G(V_1 \cup V_2 \cup \\{u\\}, E_1 \cup E_2 \cup \\{\\{v_1 \in V_1, u\\}, \\{v_2 \in V_2, u\\}\\})$.
> > >
> > > Concern2:
> > > Substantially, we think that the complexities agree with the ones ESAN and DropGNN report.
> > > ESAN reports a complexity of $O(|S|n\Delta_{max})$. To compare to our complexity $|S|$ is the number of runs $s$. ESAN assumes a constant number of layers, which is why they do not contain $l$ as a factor. Last ESAN reports the additional factor $\Delta_{max}$ since this analysis assumes nodes can send different messages to each neighbor (for example because of edge features or passing the recipient node state to the message function). Therefore, each message step increases from $O(n)$ (one message per node for all neighbors) to $O(n\Delta_{max})$ (one message per neighbor per node) . This analysis is a bit more general. We will update our analysis to possibly different messages for each edge.
> > >
> > > DropGNN reports a complexity of $O(rn)$ where $r$ is the number of runs. This model assumes constantly many layers and does not account for sending different messages to neihgbors. If we allow different messages, the complexity would be $O(rn\Delta_{max})$.
> > >
> > > In AMP, we will need to compute $\Delta_{max}$ different messages per loop iteration, resulting in $O(k \Delta_{max})$ per run of Algorithm 1 and $O(k \Delta_{max} n)$ overall. $k$ like $l$ is usually constant which gives $O(n\Delta_{max})$.
> > >
> > > Concern 4:
> > > This is a good point, added the intuition into the main text.
> > >
> > > Concern3:
> > >
> > > Thank you for asking again, there were indeed two errors in the proof. We updated the paper. The update needed an additional bit, to handle the very first message per node separately.
> > >
> > > The first error is that degree $1$ nodes that need a bit of special handling: An easy option that we take here is to add to special nodes to the graph that we connect with every degree $1$ node (which are now degree $3$) and each other. The two nodes are special in the sense that they always send state $0$, so they are transparent for sum aggregation (other aggregations will require different neural messages).
> > >
> > > The second error is that the protocol did not allow different parts of the graph to proceed at different rates (see the end of the example for illustration: $v_1$ may work on the second layer while $v_4$ is still collecting messages from the first.) This required small adjustments in the initialization and Table4.
> > >
> > > To your concern3: If we understand correctly, we have the following scenario: $v_2$ sends $m_1$ to $v_1$, which sends $m_2$ to $v_2$ and $v_3$, which both with $m_3$ and $m_4$, respectively. In the end we want to keep one message from $v_2$ and $v_3$ each. In other words, discard the second message $m_3$ from $v_2$?
> > >
> > > In principle, this should not be a problem. According to Algorithm 1, every time $v_2$ (or any node) sends a message, they can do a state update before. Then nodes can use the new state to construct the message. In the example above, $v_2$ and $v_3$ could count how many times they sent messages and include this in the message. So $m_3$ could include the information that this is the second message $v_2$ sends. On the other hand, $m_4$ would be the first message $v_3$ sends. The discard function $\rho$ could filter out any message where this count is greater than $1$. This way $v_1$ would receive $m_2$ and $m_4$ and discard $m_3$.

---

> ### Author Response · Authors · 2022-11-15
> **Response to reviewer 6FHJ 1/2**
>
> Thank your for your review.
>
> > The authors may want to provide a rigorous proof for Theorem 4.7. Some weaknesses are as follows. 1. In the proof, the authors make the assumption of GNN's width and depth, which is not mentioned in Theorem 4.7. 2. Corollary 3.1 in [1] assumes that the width of GNNs is unbounded, which is not equivalent to the assumption of a sufficiently wide GNN. 3. The authors may want to provide the lower bound of the depth of GNNs. 4 The application of Corollary 3.1 is confusing. What is the definition of Turing computable function in Corollary 3.1 in [1]? If a GNN can compute any Turing computable function, how to show that it can separate any pair of graphs?
>
> Thanks for all 4 points we hope we addressed adequately in the revised version. @1 We added those assumptions to the theorem and expanded its text a bit. @2 This is correct, the Corollary requires unbounded GNNs and thus would require an unbounded AMP. We changed the initial proof accordingly. We further discuss what bound would suffice if we do not care about every algorithm, but only graph isomorphism. In that case $O(\frac{n}{2}!)$ would be sufficiently wide. @3 we included the depth @4 Turing computable functions are those that a Turing machine can solve, graph isomorphism is one of these functions. Thus, if a GNN can solve any Turing computable function on G, it can solve graph isomorphism.
>
>
> > From Table 3, we can see that the accuracy of AMP is clearly lower than existing powerful GNNs (e.g., ESAN). The authors claim that the low accuracy is due to little investigation into AMP architectures and hyperparameters, but it is not the key reason in my opinion, as more AMP architectures with more hyperparameters are available to ensure a fair comparison for the authors.
>
> Yes, we agree that the availability of hyperparameters is not a problem. Our bottleneck is the time required to run one configuration of such parameters (due to the technical issues). Generally, we do not know yet what ranges for hyperparameters in AMP generally make sense and if general techniques such as batch-norm should be applied and how.
>
> > The authors may want to compare the time complexity and runtimes of AMP with those of synchronous GNNs.
>
> Thanks for the suggestion, we added an analysis in the general response above and in section 3.3 in the revised paper. Generally AMP and GNNs perform comparable many messages and node updates.
>
> > The authors claim that AMP can propagate messages over large distances in graphs without the corresponding theoretical analysis.
>
> It is true we discussed this claim only experimentally in the paper and had to move that analysis into the appendix for space reasons. Table 7 breaks down the accuracy by distance to the starting node. We can see that AMP-ACT and AMP-Iter in particular can classify further away nodes and also other AMP variants fail later than synchronous GNNs. We expanded the discussion of results in B.2 to give some more intuition on the results: This is not yet generalized to arbitrary problems but gives an idea why long-range messages can be better propagated in AMP:
>
> In this task, every node needs to receive one message to classify correctly (if shortest paths are not unique). Basically every other message is noise.
> * Let us look at node $v$ in a GNN that has distance $d$ to the starting node. If we use IterGNN, node $v$ needs to run at least $d$ layers before receiving the important message and terminating. In total, $v$ receives $d\cdot deg_v$ many messages where $deg_v$ is its degree.
> * In AMP, nodes discard all unimportant messages. This means that every node discards all but one message and sends a message to its neighbors exactly once. In total every node $v$ receives a total of $deg_v$ messages, independent of $d$.

---

### Author Response · Authors · 2022-12-11
**Thanks again for your reviews**

We wanted to give one more update before the discussion phase is over. We are continuing to work to make AMP faster, this is a suggestion that all of you mentioned as crucial going forward.

We would like to thank you again for you reviews and availability in the discussion, we feel the discussion helped us both in clarity, for example, improving sections 3 or 4) and prompted new ideas such as closer analyzing the effects in long range experiments.

---

### Decision · Program_Chairs · 2023-01-20

**Decision:**

Reject

**Justification For Why Not Higher Score:**

The reviewers have suggested several items that could be improved.

**Justification For Why Not Lower Score:**

NA

**Metareview: Summary, Strengths And Weaknesses:**

The paper proposes a framework for neural networks on graphs with asynchronous message passing.

* Strengths are the proposition of an interesting new idea of asynchronous message passing.
* Weaknesses are high computation cost and inaccuracy of the proposed methods, insufficient experiments, insufficient writing quality.

During the discussion period the authors could clarify several of the initial reviewer concerns while other items persisted.

The final reviewer recommendations were mixed borderline with two 6 and two 5. In the final discussion one of the 6 agreed with some of the items from one of the 5 and found significant weaknesses on the computational cost, lack of larger benchmarks, and writing. The authors also acknowledged that the method suffers from computational bottlenecks and that meaningful ranges of hyper parameter values are not yet known.

I conclude that the paper makes promising advances in an interesting direction but that it still needs to be developed.




**Summary Of Ac-Reviewer Meeting:**

NA